# A practical solution to pseudoreplication bias in single-cell studies

Kip D. Zimmerman [1,2 ✉], Mark A. Espeland[1] & Carl D. Langefeld [1,2,3 ✉]

Cells from the same individual share common genetic and environmental backgrounds and are not statistically independent; therefore, they are subsamples or pseudoreplicates. Thus, single-cell data have a hierarchical structure that many current single-cell methods do not address, leading to biased inference, highly inflated type 1 error rates, and reduced robustness and reproducibility. This includes methods that use a batch effect correction for individual as a means of accounting for within-sample correlation. Here, we document this dependence across a range of cell types and show that pseudo-bulk aggregation methods are conservative and underpowered relative to mixed models. To compute differential expression within a specific cell type across treatment groups, we propose applying generalized linear mixed models with a random effect for individual, to properly account for both zero inflation and the correlation structure among measures from cells within an individual. Finally, we provide power estimates across a range of experimental conditions to assist researchers in designing appropriately powered studies.

[1] Department of Biostatistics and Data Science, Wake Forest School of Medicine, Winston-Salem, NC, USA. [2] Center for Precision Medicine, Wake Forest School of Medicine, Winston-Salem, NC, USA. [3] Comprehensive Cancer Center, Wake Forest Baptist Medical Center, Winston-Salem, NC, USA. ✉email: kdzimmer@wakehealth.edu; clangefe@wakehealth.edu

The rapid evolution of single-cell technologies will enable novel interrogation of fundamental questions in biology, accelerating discoveries across many biological disciplines. Common fields of application for single-cell technologies include cancer, neurological disease, developmental biology, diabetes, and autoimmune disease. Thus, researchers are developing methods that leverage or account for the unique properties of single-cell RNA sequencing (scRNA-seq) data, particularly their increased sparseness and heterogeneity compared to bulk sequencing counterparts[1–3]. An important characteristic of single-cell experiments is that they use many cells from the same individual, and therefore the same genetic and environmental background. Here we empirically document correlation among measures from cells within an individual, and demonstrate how testing for differential expression analysis in scRNA-seq data within a cell type across conditions without considering this correlation — the current common practice — violates fundamental assumptions and leads to false conclusions. While differential expression analysis can be computed across all cell types, throughout this manuscript, differential expression analysis is generally considered to be computed within a specific cell type of interest (i.e., after cell clustering and cell type identification).

Proper identification of the experimental unit (i.e., the smallest observation for which independence can be assumed) for the hypothesis is critical for proper inference. Observations nested within an experimental unit are referred to as subsamples, technical replicates, or pseudoreplicates. Pseudoreplication, or subsampling, is formally defined as "the use of inferential statistics where replicates are not statistically independent"[4]. There are two types of pseudoreplication commonly occurring in single-cell experiments: simple and sacrificial. Simple pseudoreplication occurs when "samples from a single experimental unit are treated as replicates representing multiple experimental units"[4–6]. Sacrificial pseudoreplication occurs when "samples taken from each experimental unit are treated as independent replicates"[4–6]. Pseudoreplication has been addressed repeatedly in ecology, agriculture, psychology, and neuroscience and acknowledged as one of the most common statistical mistakes in scientific literature[4–9]. New technologies are particularly prone to this error. Thus, it is not surprising that, when performing a literature review prior to conducting these analyses, we found pseudoreplication to be pervasive in the single-cell literature. Properly identifying the right experimental unit, and analyzing the data accordingly, need to be urgently addressed in single-cell studies before a lack of reproducibility tarnishes the single-cell technology itself as potentially unreliable.

In this study, we simulate hierarchical single-cell expression data and evaluate the type 1 error rates and power of mixed models relative to some of the most frequently applied differential expression methods. We assess a number of commonly applied differential expression methods, but we primarily focus on the computation of a two-part hurdle model. This model explicitly accounts for the common problem of zero inflation in scRNA-seq data by simultaneously modeling the rate of expression and the positive expression mean[10]. Using the two-part hurdle model, we compute differential expression as it is most typically applied in the literature: without a random effect for individual. We then re-evaluate the two-part hurdle model's performance when computing differential expression with a random effect for individual, and after applying a batch effect correction for individual. Additionally, we examine the type 1 error control and power of aggregation (i.e., pseudo-bulk) methods, where gene expression values are averaged across all cells within an individual and the test statistic is computed on the individual means[11–13]. Aggregation methods are implemented to control for both zero-inflation and within-sample correlation, but are conservative in

unbalanced situations, which are common in single-cell data. Overall, these simulations indicate how properly accounting for the correlation structure among measures from cells within an individual will greatly increase both robustness and reproducibility, thereby leveraging the very features that make single-cell methods powerful.

## Results

**Intra-individual correlation.** We hypothesize that measures from cells from the same individual should be more (positively) correlated with each other than cells from unrelated individuals. Empirically, this appears true across a range of cell types (Fig. 1). We demonstrate this effect by estimating the pairwise correlation of cells within an individual and across any two different individuals. For a given cell type, the correlation of cells within an individual (intra-individual correlation) is always higher than the correlation of cells across individuals (inter-individual correlation). Thus, single-cell data have a hierarchical structure in which the single cells may not be mutually independent and have a study-specific correlation (e.g., exchangeable correlation within an individual). Within a cell type, cells appear to also exhibit some correlation across individuals (Fig. 1). We hypothesize this

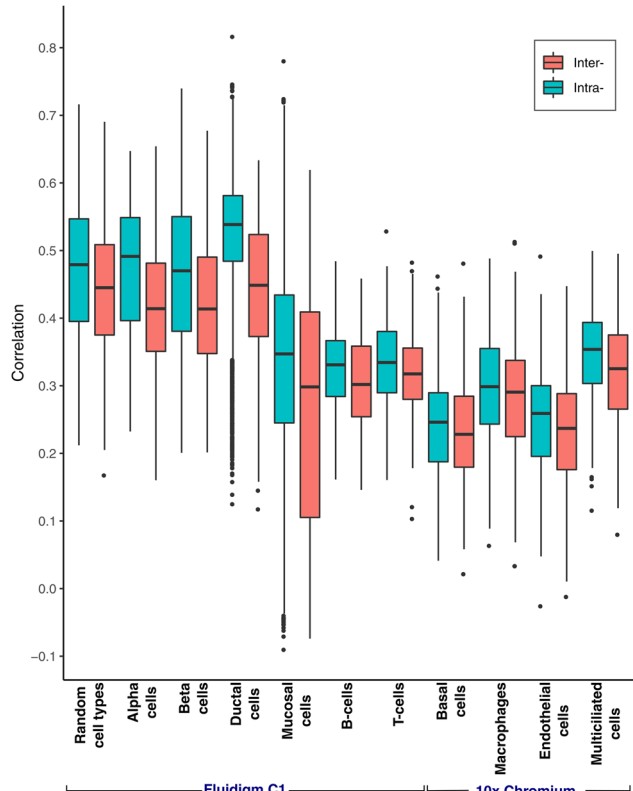

**Fig. 1 Intra-individual correlation. Intra- and inter-individual Spearman's correlations for gene expression values across ten different pancreatic cell types and a random sample of different cell types.** The respective numbers of cells and individuals used for each cell type are listed in the Methods section. Median correlation among a donor's own cells (intra-individual) is always greater than the mean correlation across individuals (inter-individual). The "random cell types" boxplots represent a random sampling of alpha, beta, and ductal cells. The center line represents the median. The lower and upper box limits represent the 25% and 75% quantiles, respectively. The whiskers extend to the largest observation within the box limit ±1.5 × interquartile range. $n = 40{,}775$ cells from 43 different individuals over four independent experiments. More details about each experiment are provided in Methods.

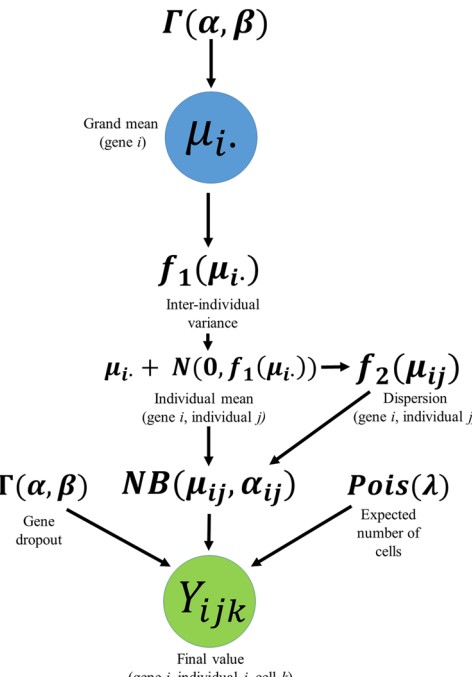

**Fig. 2 Simulation workflow.** A gamma [$\Gamma(\alpha, \beta)$] distribution was fit to the global mean transcript-per-million (TPM) value of each gene and used to obtain a grand mean, $\mu_i$ The variance of the individual-specific means (inter-individual variance) was modeled as a linear function of the grand mean, $f_1(\mu_i)$. Using a normal $N(\mu, \sigma^2)$ distribution with an expected value of zero and a variance computed by the linear relationship, $f_1(\mu_i)$, a difference in means was drawn for each individual in the simulation. This difference was summed with the grand mean to obtain an individual mean, $\mu_{ij}$. Within-sample dispersion was simulated as a logarithmic function of the inter-individual mean, $f_2(\mu_{ij})$. A Poisson ($\lambda$) distribution with a $\lambda$ equal to the expected number of cells desired for each individual was then used to obtain the count of cells per individual. The probability of dropout was estimated as a gamma distribution. For each cell assigned to an individual, a count, $Y_{ijk}$, was drawn from a negative binomial distribution.

is due to stability in functional gene expression that is needed for a cell to be classified as a specific type (e.g., T-cells need to have some consistent signals of gene expression related to their function as T-cells).

**Simulation.** We completed a simulation study that reproduced both the inter- and intra-individual variance structures estimated from real data and documented the effect of intra-individual correlation on the type 1 error rates of the most frequently used single-cell analysis tools (Fig. 2 and Supplementary Figs. 1–4). Our simulation captures some of the most important aspects of single-cell data (Supplementary Figs. 1–4) and was used to compare methods that do and do not account for the repeated observations within an experimental unit (see Methods). We varied the number of individuals and cells within an individual, and all methods considered use asymptotic approximations and admit covariates.

**Type 1 error evaluations.** The generalized linear mixed model (GLMM), employing either a tweedie distribution or a two-part hurdle model with a random effect (RE) for the individual, outperformed other methods across a variety of conditions (Table 1 and Supplementary Tables 1–4).

Among the methods that explicitly model the correlation structure, GLMM consistently better controlled for type 1 error

rate than generalized estimating equation (GEE1) models. The latter performed poorly for all numbers of subsamples until the number of independent experimental units approached 30. However, all models that explicitly model the correlation structure have more appropriate type 1 error rates than methods that do not account for lack of independence among experimental units (Table 1 and Supplementary Tables 1–4). As the number of correlated cells rose, performance of all methods that treat observations independently grew increasingly worse (Table 1 and Supplementary Tables 1–4).

One of the most heavily cited single-cell analysis tools, model-based analysis of single-cell transcriptomics (MAST), is a two-part hurdle model built to handle sparse and bimodally distributed single-cell data[10]. Although to our knowledge no publications have employed MAST to account for pseudoreplication as discussed here, Finak et al. note that MAST "can easily be extended to accommodate random effects"[10]. When implementing MAST with a random effect for individual (i.e., MAST with RE), the type 1 error rate is well-controlled. However, its type 1 error rate is just as inflated as other tools when it is not implemented with a random effect for individual. However, one suggested approach to account for within-individual correlation is the aggregation of cell-type-specific expression values within an individual by using either a sum or a mean[11–13]. Such analysis methods, as would be expected, do control for the type 1 error rate, but are conservative (Table 1 and Supplementary Tables 1–4).

Another method that could be used to account for within-sample correlations is to apply a batch effect correction method prior to differential expression, for which the batches are different individuals. Here, when we applied batch effect correction via ComBat[14] prior to differential expression analysis within a cell type, type 1 error rates markedly increased (Table 1 and Supplementary Tables 1–4).

In addition to evaluating type 1 error rates, we examined the preservation of the rank-order of results from these methods (Supplementary Table 5). We also evaluated the sensitivity (the proportion of correctly identified true positives) at varying fold changes of the two-part hurdle model when ignoring the within-individual correlation (MAST), correcting it for "batch effect" prior to differential expression (MAST ComBat), or correcting it with a random effect for individual (MAST RE) (Supplementary Fig. 5). We did not explicitly evaluate specificity (the proportion of correctly identified true negatives), which is simply computed as 1-type error. Thus, when the type 1 error rate for a method is inflated, the specificity is small. We found the highest correlations between the absolute value of the simulated-log(fold-change) and the methods that properly account for within-person correlation. The methods that do not do so maintained some semblance of rank-order, except for batch effect-corrected results (Supplementary Table 5). As expected, not properly accounting for within-person correlation leads to extremely high sensitivity with very low specificity (Supplementary Fig. 5).

**Power analyses.** We computed an extensive simulation-based power analysis to provide estimates across a wide range of experimental conditions. We used a two-part hurdle model with random effects for individuals as implemented in MAST[10]. We also computed power when expression values are averaged across cells within an individual. Increasing the number of independent experimental units (e.g., individuals) in a study is the best way to increase power to detect true differences (Fig. 3a). Empirically, when sample sizes become greater than 20, there are only marginal gains in power when more than 100 cells per individual are sampled for a particular analysis unit (i.e., computing the analysis within a single cell type of interest or across all cell types).

**Table 1 Type I error rates of some currently applied tools in single-cell analyses.** Type I error rates of ten different methods under twenty different conditions and a significance threshold of $p < 0.05$. In all, 250,000 iterations were computed to obtain an error rate for each method. The inflated type I error rates computed with mixed models at the lower number of individuals per group are a consequence of the two-part hurdle model simultaneously testing two hypotheses and an overabundance of subsampling with small sample sizes. Type I error rates are well-controlled for with mixed models and pseudo-bulk methods, while type I error rates increase with other methods as additional independent samples or more cells are added. Pseudo-bulk methods are overly conservative. Confidence intervals (95%) are included in the enlarged version of this table (Supplementary Table 1).

| $N_{ind}$ | $N_{cells}$ | Two-part hurdle | | | Tweedie | | GEE1 | Pseudo-bulk | | Tobit | Modified $t$ |
|---|---|---|---|---|---|---|---|---|---|---|---|
| | | Default | Corrected | RE | GLMM | GLM | | Mean | Sum | | |
| 5 | 50 | 0.561 | 0.637 | 0.069 | 0.082 | 0.340 | 0.114 | 0.023 | 0.035 | 0.353 | 0.400 |
| | 100 | 0.677 | 0.719 | 0.064 | 0.084 | 0.463 | 0.110 | 0.022 | 0.032 | 0.471 | 0.510 |
| | 250 | 0.798 | 0.778 | 0.066 | 0.083 | 0.609 | 0.103 | 0.023 | 0.028 | 0.628 | 0.644 |
| | 500 | 0.862 | 0.803 | 0.065 | 0.081 | 0.705 | 0.104 | 0.023 | 0.026 | 0.725 | 0.718 |
| 10 | 50 | 0.563 | 0.611 | 0.055 | 0.064 | 0.350 | 0.076 | 0.024 | 0.021 | 0.345 | 0.397 |
| | 100 | 0.689 | 0.718 | 0.053 | 0.065 | 0.462 | 0.077 | 0.024 | 0.020 | 0.470 | 0.502 |
| | 250 | 0.810 | 0.793 | 0.049 | 0.064 | 0.610 | 0.074 | 0.022 | 0.019 | 0.624 | 0.635 |
| | 500 | 0.875 | 0.827 | 0.049 | 0.061 | 0.705 | 0.073 | 0.021 | 0.018 | 0.722 | 0.717 |
| 20 | 50 | 0.562 | 0.606 | 0.051 | 0.056 | 0.344 | 0.063 | 0.024 | 0.016 | 0.343 | 0.393 |
| | 100 | 0.687 | 0.705 | 0.048 | 0.056 | 0.459 | 0.064 | 0.024 | 0.014 | 0.466 | 0.503 |
| | 250 | 0.817 | 0.805 | 0.042 | 0.058 | 0.610 | 0.060 | 0.022 | 0.011 | 0.619 | 0.637 |
| | 500 | 0.884 | 0.844 | 0.042 | 0.055 | 0.705 | 0.062 | 0.021 | 0.010 | 0.720 | 0.716 |
| 30 | 50 | 0.563 | 0.604 | 0.053 | 0.054 | 0.341 | 0.058 | 0.025 | 0.013 | 0.344 | 0.395 |
| | 100 | 0.691 | 0.698 | 0.049 | 0.056 | 0.463 | 0.058 | 0.025 | 0.012 | 0.469 | 0.504 |
| | 250 | 0.818 | 0.803 | 0.044 | 0.055 | 0.608 | 0.057 | 0.022 | 0.010 | 0.624 | 0.636 |
| | 500 | 0.886 | 0.853 | 0.041 | 0.055 | 0.707 | 0.058 | 0.022 | 0.009 | 0.719 | 0.706 |
| 40 | 50 | 0.561 | 0.602 | 0.051 | 0.054 | 0.345 | 0.055 | 0.025 | 0.013 | 0.340 | 0.393 |
| | 100 | 0.689 | 0.699 | 0.049 | 0.053 | 0.455 | 0.055 | 0.026 | 0.012 | 0.467 | 0.502 |
| | 250 | 0.820 | 0.803 | 0.044 | 0.053 | 0.607 | 0.053 | 0.022 | 0.010 | 0.622 | 0.639 |
| | 500 | 0.888 | 0.856 | 0.042 | 0.053 | 0.704 | 0.054 | 0.022 | 0.008 | 0.721 | 0.713 |

Default denotes MAST was implemented without random effects, RE denotes random effects, Corrected denotes data were batch-corrected for individual with ComBat prior to analysis without using individual as a random effect, GLM denotes generalized linear model, and GLMM denotes generalized linear mixed-effects model.
Two-part hurdle model as implemented in MAST, Tweedie distribution as implemented in "glmmTMB", GEE1 as implemented in "geepack", Pseudo-bulk averaged or summed across cells within an individual and was implemented in DESeq2, Modified $t$ as implemented in ROTS, and Tobit as implemented in Monocle.

Methods that aggregate information across cells within an individual by averaging or summing (i.e., "pseudo-bulk" methods) are only slightly underpowered relative to mixed-effects models when there is balance in the number of cells per individual. However, these are not as well powered as mixed-effects models when the number of cells per individual grow increasingly imbalanced (Supplementary Figs. 6–8).

## Discussion

Single-cell studies designed to identify differentially expressed genes rarely mention or address the correlation among cells from the same individual or experimental unit. Excellent reviews of the field and methodologic work have largely focused on challenges presented by properly classifying cell types, multimodality, dropout, and higher noise derived from biological and technical factors. However, they fail to highlight the effects of pseudoreplication. Furthermore, papers evaluating the performance of single-cell-specific tools all compute the simulations as if cells were independent[15–21]. The result is reduced reproducibility with real data, leading to the conclusion that tools built specifically to handle single-cell data do not appear to perform better than tools created for bulk data analysis[22–24].

Here, we have empirically documented the correlation among measures from cells within an individual for a few independent datasets and different cell types (Fig. 1). These findings imply that the current practice — testing for differential expression analysis across conditions in scRNA-seq data within a cell type without considering this correlation — leads to pseudoreplication.

Pseudoreplication, formally defined as "the use of inferential statistics where replicates are not statistically independent", has been addressed repeatedly in both new and well-established scientific fields[4–8]. Recently, it was acknowledged as one of the most common statistical mistakes in scientific literature[9]. Here we hope to address pseudoreplication in single-cell analyses by demonstrating what a large and long-standing body of statistical literature already confirms: applying statistical inference to replicates that are not statistically independent without properly accounting for their correlation structure will inflate type 1 error rates and lead to spurious results[4–6,9,25–28].

In our results, models that explicitly parameterized the correlation structure all showed improved type 1 error control compared to methods that did not account for the lack of independence among experimental units (Table 1 and Supplementary Tables 1–4). Furthermore, as the number of correlated cells increased, performance of all methods that treated observations independently increasingly worsened (Table 1 and Supplementary Tables 1–4).

Both the two-part hurdle mixed model and the tweedie mixed model showed type 1 error control when adjusting for individual as a random effect (i.e., MAST with RE/Tweedie GLMM), but their type 1 error rates were highly inflated when not doing so. These specific evaluations of models with and without a random effect for individual illustrate why accounting for pseudoreplication is so important. As the denominator of most statistical tests (e.g., Wald test) is a function of the variance, not accounting for the positive correlation among sampling units underestimates the true standard error and leads to false positives[27,28]. In addition,

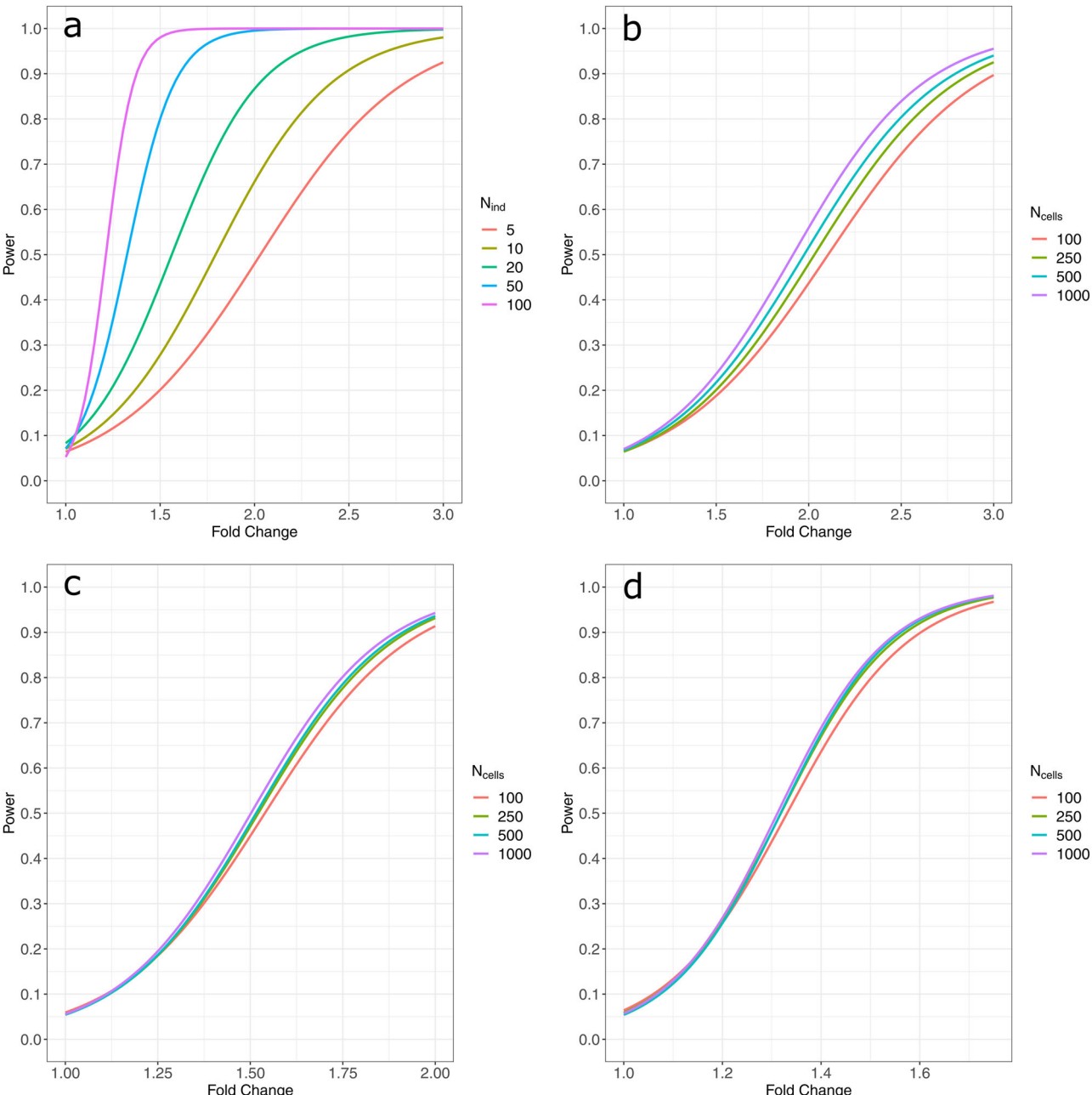

**Fig. 3 Power calculations using MAST with a random effect for the individual.** Power curves for various, but likely, single-cell scenarios using MAST with a random effect for individual. Fold-change is simulated by multiplying the global mean gene expression values by the fold-change value for one group. All power is computed at $\alpha = 0.05$. **a** Differences in power when sample sizes range between 5 individuals per group to 100 when the number of cells per individual is held constant at 250. **b-d** Differences in power when increasing the number of cells per individual (100, 250, 500, 1000) for 10, 20, and 50 individuals per group, respectively. Additional power curves are supplied in the supplementary material.

treating each cell as independent inflates the test degrees of freedom, making it easier to falsely reject the null hypothesis (type 1 error). This approach leads to apparently greater sensitivity among methods that do not properly account for pseudoreplication (Supplementary Fig. 5). However, the type 1 error rates (and, thereby, specificity) indicate that, while these methods capture most results at the lower fold-changes, they also capture a large proportion of false positives. Too many false positive results can mask true associations, especially when multiple comparison procedures such as false discovery rate are applied. In combination, this will adversely affect downstream analyses (pathway analysis), robustness, and reproducibility — all increasing the cost of science.

One potential method to remove inter-individual differences prior to analysis is to apply batch effect correction prior to differential expression analysis, where the batches are individuals. Using these techniques to correct for intra-individual correlation should, in fact, be used more often before cell-type clustering, to more accurately identify cell types upstream of testing for differentially expressed cell-type marker genes with mixed models. However, when using these methods prior to differential expression analysis within a cell type, they show markedly increased type 1 error rates (Table 1 and Supplementary Tables 1–4). In these analyses, we used the batch effect correction tool, ComBat[14]. Many other batch effect correction tools are available. However, the underlying concept common to all is that

they should not be applied to account for within-person correlation when estimating the variance used in testing the hypothesis of differential expression. This is primarily because regressing out the person-specific effect as a batch effect and subsequently analyzing each cell as an independent observation will underestimate the overall variance by removing inter-individual differences while maintaining an inappropriately large number of degrees of freedom when treating cells as if they are independent. In addition, applying a batch effect correction in this manner will greatly disturb the rank-order of results (Supplementary Table 5). However, batch correction methods are valuable when used as designed.

Another method recommended before analysis is to aggregate individual gene expression values within a person by summing or averaging them[11–13]. Such approaches, labeled "pseudo-bulk" techniques, can be appropriate ways of handling the correlation structure, but averaging within-subject measurements reduces the number of data points and loses information. The fewer data points and decreased confidence in the estimated population mean make this a conservative approach when the number of cells per individual are imbalanced. Averaging within-subject measurements ignores within-subject variability. Furthermore, when within-subject variability is large with respect to between-subject variability, averaging within-subject measurements fails to estimate the true data variability[26].

Overall, mixed-effects models lead to the most accurate results when analyzing data with a hierarchical structure[6,25,26]. As we demonstrate here, aggregating values across cells from the same experimental unit will actually lead to an increased type 2 error rate and decreased power (Table 1 and Supplementary Figs. 6–8). This is due to an overestimation of the mean-square error relative to mixed-effects models, particularly when imbalance exists and the intra-individual variance is larger than the inter-individual variance, as appears to be typical with scRNA-seq data[25,26]. In imbalanced situations, "pseudo-bulk" methods also cause cells from individuals with fewer cells to be more heavily weighted, where mixed models have consistent estimators and do not require balanced data[29]. Taking the mean across all cells within an individual will also cause problems when used together with tools that require integers for the input. In addition, taking the mean in this situation generates heterogeneity in certainty of the estimates of the mean — that is, the data are not independent and identically distributed. Errors-in-variables regression should be explored as a means of accounting for this heterogeneity[30].

The two-part hurdle model also has the ability to test for differences in both the proportion of zeros and the magnitude of effect across treatment groups separately. In some scenarios, combining the data will cause a significant difference in the magnitude of effect to be washed out by a significant difference in the proportion of zeros or vice versa (i.e., Simpson's paradox). Such scenarios occur infrequently in our simulated data, which holds the probability of zeros constant across treatment groups, but such scenarios may be more common in real data. "Pseudo-bulk" techniques may control the type 1 error rates and may help account for zero-inflation; however, we recommend mixed-effects models based on long-standing statistical justifications for the analysis of subsamples, including increased power. This is particularly so in scenarios with heavy imbalance and where within-subject variability is large compared to between-subject variability.

Among the methods that explicitly model the correlation structure, GLMMs consistently better controlled for type 1 error rate than GEE1 models. Here, GEE1 exhibited elevated type 1 error rates for any number of subsamples until the number of independent experimental units approached 30. When the number of experimental units was small, the GEE1 sandwich estimator of the variance provided standard errors that were too small and therefore inflated the type 1 error rate[31,32].

Here, we emphasize the two-part hurdle mixed model, implementable in MAST, as an already well-established tool in the field. We demonstrate that this implementation performs exceptionally well when adjusting for individual as a random effect[10]. MAST with RE is testing a two-part hypothesis that the other tools are not directly testing. The discrete and continuous components being tested fit together — meaning that higher mean expression will generally correlate with a higher proportion of expressing cells, but assuming that the two will always relate is incorrect. There will be specific instances in the simulated data when the inter-individual means are not significantly different, but the proportion of cell dropout is significantly different (even though the probability of dropout for any one gene across cells is held constant) and is driving a significant result. This will be particularly true with smaller sample sizes, and may contribute to the slightly elevated type 1 error rates with smaller sample sizes and cell counts.

While we recommend computing differential expression analysis using MAST with RE, alternative methods include using MAST with fixed-effects for individual, a tweedie GLMM, or permutation testing. Accounting for the within-sample correlation with a fixed-effect term for individual, will have a slight difference in interpretation, but should be considered an alternative option to random effects — particularly when the number of independent experimental units is modest[33]. To not violate the exchangeability assumption, permutation methods must randomize at the independent experimental unit (e.g., individual) and properly account for covariates (i.e., conditional permutation). The tweedie GLMM method, which was selected for its distributional flexibility that can account for zero inflation, could be implemented using the "glmmTMB" R-package[34], but other mixed models could also be applied using a more appropriate distribution if, for example, the observed data do not exhibit zero inflation. However, none of these alternative approaches explicitly incorporates some single-cell-specific concepts implemented in MAST (e.g., cellular detection rate).

We computed power analyses for a variety of sample sizes and cell amounts. Empirically, we demonstrated that increasing the amount of cells captured per sample returns very little gain in power after 100 cells per individual in most scenarios, particularly after sample sizes increase (Fig. 3 and Supplementary Figs. 9–11). Instead, we suggest that increasing sample size is the most efficient way to improve power (Fig. 3a). Increasing the number of cells per individual does provide more precision in the estimate for an individual. However, it has limited effects on the power for detecting differences across individuals, such as differences among treatments applied to individuals (i.e., case/control studies). Estimating power with more than 1000 cells per individual is computationally expensive; run times for the random effects models are significantly higher than the other tools (Supplementary Table 6). Because using thousands of cells per individual is not atypical for single-cell experiments, tools that account for the correlation structure when analyzing these data need to be further developed to increase computational efficiency (e.g., parallel code, use of GPU). In addition, while mixed models will compute for only two individuals per treatment group, many genes will fail due to complete separation when the sample size is so small, especially when the number of cells per individual is also small (<50).

Most papers compare cells across very few individuals, sometimes even a single case and control (simple pseudoreplication). In the former case, the estimate of the inter-individual variance is possible, but has wide bounds on parameter confidence intervals; in the latter case the variance is not estimable from the data.

Simulations indicate that most published studies are under-powered (Fig. 3 and Supplementary Figs. 9–11).

Most single-cell papers show a deep understanding of the underlying biology and conduct otherwise very informative experiments, appropriately published in high-visibility journals. However, our type 1 error and power simulations document that many such studies are missing important true effects while reporting too many false positive effects generated via pseudoreplication.

As single-cell technology continues to evolve and costs decrease, scientists need to be aware of this issue to improve study design and avoid proliferation of irreproducible results. Our results encourage the use of mixed models, such as the two-part hurdle model with a random effect (e.g., as implemented in MAST with RE), as a way to account for repeated observations from an individual while being able to adjust for covariates at the individual level and, if appropriate, at the individual cell level. Additional random effects, such as sampling time, may also be included[35]. Our extensive simulation study provides valuable information for understanding the power of specific designs and can be used in grant reviews as one justification of the design and analyses employed. A limitation of our simulation engine here is that it was modeled using plate-based data. However, we demonstrate that droplet-based single-cell data also contain a hierarchical correlation structure (Fig. 1). Although our focus here is on hypothesis testing for finding differentially expressed genes within a cell type across conditions, the concept is applicable when comparing expression patterns between cell types and is broadly appropriate for all single-cell sequencing technologies such as proteomics, metabolomics, and epigenetics. In addition, there are numerous fields such as cancer, neurological disease, developmental biology, diabetes, and autoimmune disease, in which these results apply.

## Methods

**Literature review.** A PubMed search in January 2019 for the keywords "single-cell differential expression" returned 251 articles published in the last 3 years; these were subsequently sorted and filtered by each of their abstracts. Many of the returned articles were associated with bulk RNA sequencing or completely irrelevant to differential expression analyses in single cells and were therefore eliminated. Of the 251 original hits, 76 were deemed appropriate for further consideration. Of those, 10 were reviews, 36 were methods papers, and 30 were implementation papers. Each of the methods and implementation articles was thoroughly reviewed along with its number of citations, date of publication, and any other pertinent information, such as number of independent samples, tools used, or number of cells captured.

**Intra- and inter-correlation analyses.** We made pairwise comparisons between all cells of interest to compute intra- and inter-individual correlations. Genes were removed if the average transcript-per-million (TPM) value was equal to zero. To control for the correlation structure between genes, genes were sampled one at a time, and any genes with a Spearman's correlation coefficient >0.25 relative to the gene that was drawn were subsequently trimmed from the dataset. This step was repeated until either no more uncorrelated genes remained or a total of 500 uncorrelated genes were obtained, whichever happened first. For intra-individual correlations, Spearman's correlation was computed for all possible pairs of cells within an individual. For inter-individual correlations, Spearman's correlation was computed for all possible pairs of cells from a random draw of one cell from each individual. We computed 1000 draws. To compute the correlation structure across multiple cell types, intra-individual correlations were assessed by repeatedly drawing one cell per cell type within an individual and computing all pairwise correlations. Inter-individual correlations were assessed by dividing the data into a balanced set of observations, with 10 cells of each of the three main cell types retained for each individual. The intra- and inter-individual correlations and their means were examined for differences (Fig. 1). The measures were compared in ten different cell types across four different single-cell studies. These studies are all publicly available (accession numbers GSE81861, GSE72056, E-MTAB-5061, and EGAS00001004082). The GSE81861 dataset contains 161 normal mucosal cells from 6 individuals and were sequenced on Fluidigm's C1. All individuals were patients with colorectal cancer, but tissues were taken from healthy mucosa. The GSE72056 data were also sequenced on Fluidigm's C1. The dataset used here contains 337 B-cells from tumor tissues of 11 individuals (all with melanoma) and

1186 T-cells from tumor tissues of 17 individuals (also all with melanoma). E-MTAB-5061 data were sequenced using the Smart-Seq2 protocol and contains pancreatic cells taken from 10 individuals, 6 healthy controls, and 4 with type 2 diabetes. Here, only data from 886 alpha cells, 270 beta cells, and 336 ductal cells were used. The EGAS00001004082 dataset contains 77,969 cells sampled from the respiratory tract of 10 healthy donors. The single-cell capture of these data was carried out using the 10X Genomics Chromium device (3′ V2). For these analyses, we used 24,138 basal cells, 2722 endothelial cells, 2417 macrophages, and 8322 multiciliated cells. Cell type designations were as given by the authors of these studies. More details are provided in their respective papers[36–39].

**Simulation engine.** A simulation engine was designed to simulate independent genes to approximate the hierarchical structure of real data by empirically estimating the range of parameters (i.e., grand mean of the TPM values, within-sample variance, between-sample variance, relationship between the grand mean and dispersion, and dropout) that define the observed distribution of TPM values for a gene. To estimate these parameters, genes were pruned to a set of uncorrelated genes that captured the most representative patterns of detectable TPM values, without the resulting parameter estimates being primarily driven by dropout. Specifically, genes were sequentially sampled one at a time; any other genes with transcript abundances that correlated (Spearman's correlation coefficient >0.25) with the gene were removed. To estimate the grand means independently from the hierarchical correlation structure, we estimated the grand means by sampling one cell from each individual and computing the mean TPM value 1000 times. The mean of each of those means was used to approximate the grand mean.

To approximate the variance of the within-sample means (inter-individual variance), the means of all non-zero TPM values were computed across all cells within each individual and the variance between those values was subsequently computed. To estimate the within-sample dispersion values, the non-zero TPM values were first used to compute a within-sample variance and within-sample mean. Consistent with the classical definition of the negative binomial distribution's dispersion parameter, the within-sample dispersion parameter was then computed as:

$$\alpha_{ij} = \frac{\mu_{ij}^2}{\sigma_{ij}^2} - \mu_{ij} \qquad (1)$$

where $\alpha_{ij}$ represents the dispersion parameter, $\mu_{ij}$ represents the within-sample mean, and $\sigma_{ij}^2$ represents the within-sample variance for gene $i$ and individual $j$.

The grand means and variances were computed empirically from the TPM values previously reported in six different cell types across three different single-cell studies[36–38]. Once consistent patterns were identified across cell types, alpha cells from the pancreatic cell dataset were used as the model data for our simulation. A gamma distribution was fit to the global mean of the TPM values of each gene using maximum-likelihood estimation. For each independently simulated gene $i$, a random value was sampled from this gamma distribution to obtain a grand mean, $\mu_i$. The variance of the within-sample means (inter-individual variance) was modeled as a linear function of the grand means, $f_1(\mu_i)$ and the within-sample dispersion (intra-individual variance) was estimated as a logarithmic function of the within-sample means, $f_2(\mu_i)$. The probability of dropout was estimated independently as a gamma distribution (Fig. 2). Using a normal distribution with an expected value of zero and a variance computed by the first linear relationship, $f_1(\mu_i)$, a difference in means was drawn for each of the individuals $j$ in the simulation. This difference was summed with the grand mean to obtain an individual mean, $\mu_{ij}$.

Three different methods were used to simulate the number of cells per individual. To simulate scenarios where each individual had an identical number of cells, the number of cells desired for each individual was fixed at a constant value. To simulate scenarios where the number of cells per individual demonstrated slight imbalance, a Poisson distribution with a λ equal to the expected number of cells desired for each individual was then used to obtain the count of cells for each individual. To simulate scenarios where the number of cells per individual demonstrated greater imbalance, the number of cells per individual was modeled as a negative binomial random variable with a mean equal to the expected number of cells and a dispersion parameter of one. For each gene $i$ and cell $k$ assigned to an individual $j$, a read count value, $Y_{ijk}$, was drawn from a negative binomial distribution with an expected value equal to the individual's assigned read count value, $\mu_{ij}$, and a dispersion parameter, $\alpha_{ij}$, computed by the logarithmic function of the grand mean $f_2(\mu_i)$. Along with the distributions of the primary parameters of interest, we made tSNE plots of the simulated data to assess how realistic the simulated data appeared (Supplementary Figs. 1–4).

**Evaluation of type 1 error.** To estimate type 1 error rate, we simulated TPM values for an individual gene 250,000 times for each simulation condition. We varied the simulation conditions by the number of individuals per treatment group and the number of cells per individual. For each iteration, the number of individuals per treatment group was fixed at either 5, 10, 20, 30, or 40. For each iteration at a fixed number of individuals per treatment group, the number of cells per individual was drawn from a Poisson distribution with either a λ of 50, 100, 250, or 500.

The type 1 error rate for differential expression testing was computed for each of the ten methods evaluated in this manuscript. For each, the number of results that met our significance threshold were counted and the type 1 error computed as the proportion of significant results. Type 1 error was computed for a tweedie mixed-effects model[34], MAST[10], MAST with random effects[10], MAST with a batch effect correction using ComBat[10,14], DESeq2 with aggregation methods ("Pseudo-bulk" summing and averaging)[11–13,40], Monocle[41], ROTS[42], Tweedie GLM[34], and a GEE1 with a Gaussian link and exchangeable correlation[43]. MAST was implemented with and without the use of a random effect for individual, and the remaining single-cell tools were implemented exactly as their vignettes instruct. GEE1 with exchangeable correlation was implemented to compare its performance to the mixed-effects model, particularly where the numbers of donors were low. Except for DESeq2 and ROTS, which both require raw counts, all methods were computed on the $\log(x+1)$ transformed gene expression matrix. To control for any differences in library size that DESeq2 might be falsely estimating, we fixed all of DESeq2's size factors to one. Type 1 errors were computed using significance thresholds of 0.05, 0.01, 0.001, and 0.0001 (Table 1 and Supplementary Tables 1–4).

MAST models a $\log(x+1)$ transformed gene expression matrix as a two-part generalized regression model[10]. As in Finak et al.[10], the addition of random effects for differences among persons is:

$$\text{logit}(\text{Pr}(Z_{ki}=1|X_k)) = X_k\beta_i + W_k\gamma_j \tag{2}$$

$$\text{Pr}(Y_{ki}=y|Z_{ki}=1) = N(X_k\beta_i + W_k\gamma_j, \sigma_i^2) \tag{3}$$

where $Y_{ig}$ is the expression level for gene $i$ and cell $k$, $Z_{ki}$ is an indicator for whether gene $i$ is expressed in cell $k$, $X_k$ contains the predictor variables for each cell $k$, and $W_k$ is the design matrix for the random effects of each cell $k$ belonging to each individual $j$ (i.e., the random complement to the fixed $X_k$). $\beta_i$ represents the vector of fixed-effects regression coefficients and $\gamma_j$ represents the vector of random effects (i.e., the random complement to the fixed $\beta_i$). $\gamma_j$ is distributed normally with a mean of zero and variance $\sigma_{\gamma_k}^2$. To obtain a single result for each gene, the likelihood ratio or Wald test results from each of the two components are summed and the corresponding degrees of freedom for each component are added[10]. These tests have asymptotic $\chi^2$ null distributions; they can be summed and remain asymptotically $\chi^2$ because $Z_i$ and $Y_i$ are defined conditionally independent for each gene[10].

**Evaluation of rank-order preservation**. To approximate how well rank-order was preserved across each of the ten methods evaluated, we simulated TPM values for an individual gene 2000 times with random fold-changes between 0 and 4. The number of individuals per treatment group was fixed at 30 and the number of cells per individual was fixed at 100. Fold-change was drawn from a uniform distribution with a minimum equal to 0 and a maximum equal to 4. Genes were retained along with their fold-change information to evaluate rank-order correlation and to complete the sensitivity analysis. The genes simulated under the null for the type 1 error rate calculations were used to estimate specificity (1-type error) for each method. With each of the different methods, $p$-values were computed and were ranked alongside the absolute value of the simulated-log(fold-change) values. Spearman's rank-correlation coefficients were computed between each of the different methods.

**Evaluation of power**. To estimate the power of the respective tests, TPM values were simulated for an individual gene 1000 times for each incremental fold-change of 0.01 between 1 and 5. Here, fold-change is a constant that was multiplied by the global mean gene expression values to spike the expression of those genes in the desired treatment group. The direction of the fold-change was simulated with a Bernoulli distribution with a probability of 0.5 to allow the direction of effect to vary equally between treatment groups. We varied the simulation conditions by the number of individuals per treatment group and the number of cells per individual. For the datasets used to compare the performance of mixed models with pseudo-bulk methods dependent on the degree of cellular imbalance between individuals, the number of cells per individual were either fixed at an exact number, allowed to vary under a Poisson distribution with a $\lambda$ equal to the expected number of cells, or allowed to vary under a negative binomial distribution with an overdispersion parameter equal to 1.

Using the two-part hurdle model with a random effect for individual, we computed power curves to estimate how well this method functions with varying numbers and ratios of cells and individuals. When evaluating the power of the pseudo-bulk methods, the size factors for DESeq2 were forced to one to keep DESeq2's normalization from normalizing out the simulated effects. Power was computed for fold changes between 1 and 5 (Supplementary Figs. 9–11).

**Reporting summary**. Further information on research design is available in the Nature Research Reporting Summary linked to this article.

## Data availability
All data are publicly available. Two of the datasets are available on NCBI's Gene Expression Omnibus under the accession numbers GSE81861[36] and GSE72056[37]. A third dataset is hosted on EMBL-EBI's ArrayExpress under the accession number E-MTAB-5061[38]. The fourth dataset is hosted on EMBL-EBI's European Genome-phenome Archive under the accession number EGAS00001004082[39].

## Code availability
Data were simulated in R-3.5.1. All of the code for the simulations and the evaluation of intra- and inter-individual correlation structure is available on GitHub at https://github.com/kdzimm/PseudoreplicationPaper. This base code was modified to run type 1 error and power analyses in parallel on Wake Forest's High Performance Computing Cluster, DEMON.

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

## Acknowledgements
We thank T. D. Howard, L. D. Miller, and M. Olivier (all from Wake Forest School of Medicine) for critical review of this content. This work was supported by the Wake Forest Center for Public Health Genomics and grant U01 NS036695 (Co-PI Langefeld) from NIH and by the Cancer Center Support Grant from the National Cancer Institute to the Comprehensive Cancer Center of Wake Forest Baptist Medical Center (P30CA012197).

## Author contributions
C.D.L. and K.D.Z. conceived the study together. K.D.Z. implemented simulations and analyses with guidance from C.D.L and M.A.E.; K.D.Z. wrote the original draft and reviewed and edited it with M.A.E. and C.D.L.

## Competing interests
The authors declare no competing interests.
