## [Peer Review File. · Nature Communications]

Reviewers' comments:

Reviewer #1 (Remarks to the Author):

Zimmerman and colleagues discuss the issue of pseudoreplication in single-cell differential expression analyses (cells from the same individual/multiple individuals being treated as independent replicates) and suggests to use mixed models for inference. The paper tackles a timely and important topic, but the exposition would benefit from framing the precise problem the authors are addressing a bit more clearly and providing a more thorough evaluation of the data used to draw the conclusions.

1. The term 'differential expression' can be used for different types of analyses in single-cell data (e.g., comparing expression patterns between cell types and comparing expression patterns within a cell type between sample conditions), and it would be helpful to the reader if the authors could elaborate explicitly on which one(s) they are considering.

2. Several previous publications evaluating mixed models for scRNA-seq differential expression problems [1,2] have suggested to instead use pseudo-bulk samples (aggregating the abundances across the cells from the same experimental unit). Such approaches would be very relevant to include also in this manuscript, for comparison.

3. In the discussion of Fig. 1, the authors speculate about the reason for the positive correlation between cells from different samples, within a cell type. It would be interesting to include also the correlation between cells from different cell types.

4. What is the rationale behind the distributional choices made in the simulation setup? Specifically, what is the motivation for sampling the final TPM value from a normal distribution (and what is done if the sampled value is negative)?

5. In general, it is not clear whether the simulated data recapitulates important aspects of real data (and if so, whether it is more representative of data from full-length read count protocols or UMI counts from droplet protocols), including whether the correlation structures shown in Fig. 1 can be recapitulated in the simulated data. A tSNE plot is not sufficient in this respect, since it shows only the relationships among the cells, not the characteristics of the simulated abundances. Showing that the data is representative of real scRNA-seq data is important in order to know whether the conclusions are likely to generalize.

6. The authors note that DESeq2 failed in many cases because of the normalization method. It is worth noting that DESeq2 implements also another normalization option ('poscounts') which is applicable in situations with sparse count matrices.

7. On p.6, it is written that "A particularly noteworthy approach that has been suggested to account for the within-individual correlation is applying a batch effect correction method", but there is no reference to where this claim was made.

8. How is the performance of the mixed models affected by imbalances in the number of cells between different individuals?

9. It's not clear where the results of the extensive literature review (described in the Methods section) are shown.

10. Please provide more detailed information about the data sets used to generate Fig. 1 (only accession numbers are given in the Methods section), such as the technology that was used to generate them, and explain why these data sets were selected.

[1] Lun & Marioni: Overcoming confounding plate effects in differential expression analyses of single-cell RNA-seq data. *Biostatistics* (2017)

[2] Crowell et al: On the discovery of population-specific state transitions from multi-sample multi-condition single-cell RNA sequencing data. *bioRxiv* (2019)

Reviewer #2 (Remarks to the Author):

In this paper, Zimmerman and authors examine between cell correlations within individuals in single cell data. The positive correlation observed between cells within an individual is widely accepted in the single cell field, but is usually overlooked in analysis. The authors suggest the use of mixed effects models to improve statistical testing in single cell data.

I think that this paper attempts to address an interesting and often over-looked problem in single cell data analysis. However, I have several concerns about the paper:

1) There is no evidence that the simulation that the authors propose replicates real single cell data. Several simulations have already been proposed for single cell data, as the authors themselves cite in references 12-18, but no comparison has been performed to show that the authors' simulation is "better" than other methods, or more similar to real data. I am also not convinced about the model assumptions and how the TPMs have been generated. I question whether TPMs are indeed normally distributed. Perhaps on a log scale they tend to be normally distributed, but TPMs are highly skewed data.

2) An important reference that is missing is Crowell, H.L., Sonesson, C., Germain, P.L., Calini, D., Collin, L., Raposo, C., Malhotra, D. and Robinson, M., 2019. On the discovery of population-specific state transitions from multi-sample multi-condition single-cell RNA sequencing data. *BioRxiv*, p.713412.

- The authors perform an extensive benchmarking study comparing statistical models (including mixed models, as well as MAST) for determining differential gene expression between conditions with multiple individuals in single cell RNA-seq data.
- They provide software called muscat which provides a simulation platform for multi-sample scRNA-seq data.
- It is not clear to me what novelty this paper by Zimmerman adds to what has been done in Crowell et al (2019).

3) Recently in Crowell et al (2019) as well as in Lun and Marioni (2017) the use of "pseudobulk" samples have been proposed to improve variance estimates. This entails summing the counts of the cells in each cell type per individual and performing statistical tests for differential expression between groups of interest, using already established models such as negative binomial generalised linear models (e.g. using DESeq2 or edgeR), or using normal based models on log-counts such as limma (Ritchie et al, 2015). Performing analysis on pseudobulk samples has the added benefit of substantially reducing the datasets making them quicker and easier to analyse. The authors have not mentioned this strategy, and it should be considered as it is much more computationally tractable for larger single cell datasets, which are becoming increasingly common. I must pose the question, that as datasets continue to get larger with hundreds of thousands of cells, will fitting mixed models be practical?

4) There is no detail about the dataset(s) in the main paper in Figure 1. Is this the only dataset that was analysed for the purpose of showing strong intra-individual correlation? Are the data generated from droplet-based technology, or plate based? Are the datasets generated on different platforms? There is no other visualisation of this data, such as tSNE or UMAP plots, making it difficult to understand the underlying relationships between the cells. Are cells within a cell type more correlated between individuals, than cells within an individual but between cell types? In a complex tissue such as heart or kidney, the correlation structure could be quite different compared to a more homogenous population of cells. There is simply not enough data shown to make broad conclusions about the importance of intra- and inter-individual correlations between cells, and the interplay between tissue complexity, cell types and individuals.

Minor comments:

1) The paper lacks a coherent structure. This makes the paper more difficult to read, as well as assess the novelty of the proposed solution by the authors. For example, having sections such as

simulation study, proposed statistical model, comparison with other methods in the field would be helpful. A lot of detail is in the supplementary and I would prefer more detail in the paper.

2) The authors don't state the null hypothesis that is being tested, and often refer to "differential expression", or "finding differentially expressed genes". I gather that the authors are referring to a very specific scenario of testing for differentially expressed genes between groups, such as treatment/control, after clustering and cell type identification has been performed. There is no detail regarding whether the testing is performed within cell types. There is also no discussion about the effect of intra-individual correlation on finding marker genes between clusters, which is another often ignored problem in the field.

3) In the Supplementary methods the authors state: "DESeq2 requires integers and at least one gene without a zero value to compute its normalization, so as the number of samples and cells increased, the likelihood of if computing greatly decreased." I would like to point out that edgeR (McCarthy, Chen and Smyth 2012) is another package designed for bulk RNA-seq, that CAN handle TPM/RPKM and doesn't have the same issue as DESeq2's normalization.

References:

Aaron T. L. Lun, John C. Marioni, Overcoming confounding plate effects in differential expression analyses of single-cell RNA-seq data, *Biostatistics*, Volume 18, Issue 3, July 2017, Pages 451–464, <https://doi.org/10.1093/biostatistics/kxw055>

Crowell, H.L., Sonesson, C., Germain, P.L., Calini, D., Collin, L., Raposo, C., Malhotra, D. and Robinson, M., 2019. On the discovery of population-specific state transitions from multi-sample multi-condition single-cell RNA sequencing data. *BioRxiv*, p.713412.

Matthew E. Ritchie, Belinda Phipson, Di Wu, Yifang Hu, Charity W. Law, Wei Shi, Gordon K. Smyth, *limma* powers differential expression analyses for RNA-sequencing and microarray studies, *Nucleic Acids Research*, Volume 43, Issue 7, 20 April 2015, Page e47, <https://doi.org/10.1093/nar/gkv007>

McCarthy DJ, Chen Y, Smyth GK (2012). "Differential expression analysis of multifactor RNA-Seq experiments with respect to biological variation." *Nucleic Acids Research*, 40(10), 4288-4297. doi: 10.1093/nar/gks042.

We thank the reviewers for their time and thoughtful review of our manuscript. After reworking some of the key components of our manuscript, we feel the revision is much stronger.

Particularly, we thank both reviewers for presenting us with the recently submitted *BioRxiv* manuscript implementing ‘pseudo-bulk’ methods and for taking a critical look at our simulation methods. The largest changes implemented in our manuscript are bulleted below, and more comprehensive responses to each of the reviewer’s comments and requests are detailed in the following pages.

The largest and most critical changes to our manuscript included:

- We examined how well our simulation recapitulated real data, and we were unsatisfied with the result. We updated our methods for simulation (Fig. 2) and reassessed how well the new simulation recapitulated real data. This was done iteratively until we were satisfied with the performance of our simulation relative to a number of key parameters in real data (Fig. S1-S4). Some critical changes to the simulation included using a negative binomial probability mass function rather than a normal distribution and removing the gene-gene correlation structure prior to estimating parameters. Hereafter, we will follow the convention in the literature by denoting the negative binomial probability mass function as a distribution.
- We used the newly modified simulation to re-run all of our type 1 error and power calculations.
- Per the reviewers’ recommendations, we included ‘pseudobulk’ methods in all of our type 1 error and power calculations to compare their performance to mixed models. Here, we demonstrate that mixed effects models are preferable to aggregation methods; albeit they do require more computation time.
- We cited and discussed the costs and benefits of implementing ‘pseudobulk’ methods compared to mixed models. This discussion includes the findings of a separate

manuscript that mathematically and empirically demonstrates why mixed models will consistently outperform methods that ignore the correlation structure as well as methods that aggregate information within an independent experimental unit. This is a general result of mixed models and not subject matter dependent.

Again, we thank the reviewers for their time and efforts.

Sincerely,

Carl D. Langefeld, Ph.D.
Professor, Department of Biostatistics and Data
Science
Division of Public Health Sciences
Wake Forest School of Medicine
Medical Center Boulevard, WC 2321
Winston-Salem, NC 27157-1063

Kip D. Zimmerman, B.A.S.
PhD Candidate, Molecular Genetics and
Genomics
Wake Forest School of Medicine
Medical Center Boulevard, WC 2321
Winston-Salem, NC 27157-1063

Reviewer #1 (Remarks to the Author):

Zimmerman and colleagues discuss the issue of pseudoreplication in single-cell differential expression analyses (cells from the same individual/multiple individuals being treated as independent replicates) and suggests to use mixed models for inference. The paper tackles a timely and important topic, but the exposition would benefit from framing the precise problem the authors are addressing a bit more clearly and providing a more thorough evaluation of the data used to draw the conclusions.

1. The term 'differential expression' can be used for different types of analyses in single-cell data (e.g., comparing expression patterns between cell types and comparing expression patterns within a cell type between sample conditions), and it would be helpful to the reader if the authors could elaborate explicitly on which one(s) they are considering.

Comparing expression patterns within a cell type between sample conditions is the question of interest being addressed here. We have clarified this in our manuscript (pg. 2 & 14), but comparing expression patterns between cell types is also relevant here as researchers commonly treat cells independently in those scenarios as well.

2. Several previous publications evaluating mixed models for scRNA-seq differential expression problems [1,2] have suggested to instead use pseudo-bulk samples (aggregating the abundances across the cells from the same experimental unit). Such approaches would be very relevant to include also in this manuscript, for comparison.

We thank the reviewer for suggesting the comparison to pseudo-bulk sample methods. We now cite 'pseudo-bulk' techniques in our manuscript (pg. 11), and we have compared their performance with that of mixed models (Table 1, Tables S1-S4, Figure 3). Reviewer 2 also referenced this approach, and additional comments regarding pseudo-bulk approaches can be seen in item 2) below. Summing or averaging across all cells within an individual is a loss of information and that while it will most certainly control the type 1 error rate (Table 1), it will reduce power and decrease one's ability to detect true differences (Figure 3). In situations of highly imbalanced data, 'pseudo-bulk' methods would cause aggregated values from individuals with fewer cells to have comparable weight to the aggregated values from individuals with more cells unless weighted in some manner (e.g., sample size, variance of the mean). In particular, summing across cells will be heavily biased by the number of cells present in each

individual (i.e., cells expressing an identical amount in two individuals with a large difference in the number of cells will be identified as DE). The precision of the mean for an individual will vary, potentially substantially, with large differences in cell counts per individual, and these analyses of those means should employ error in variable models. Beyond what is simulated here, there are very particular situations where mixed-models will identify differences that ‘pseudo-bulk’ techniques will not:

- 1) Differences where the means are the same, but the expression patterns are very different (i.e., half low/half high vs all cells in the middle)
- 2) Differences in the proportion of cells expressing versus not expressing like the hurdle model in MAST is designed to detect.
- 3) Differences where the mean difference in expression is very small, but the variance is very tight (‘pseudo-bulk’ techniques discard the intra-individual variance).

3. In the discussion of Fig. 1, the authors speculate about the reason for the positive correlation between cells from different samples, within a cell type. It would be interesting to include also the correlation between cells from different cell types.

We have added a boxplot for the inter- and intra-individual correlations of the expression across cells of all pancreatic cell types (Figure 1). The reason for using this dataset for this component is because it is the only dataset among those we were able to download with multiple cell subtypes having cell numbers greater than 100. We have also changed the text (pg. 4) to focus more on the zero-inflation component of the correlation, since the correlation among cells from different cell types is only slightly smaller than the correlation within particular cell types from the same dataset (alpha, ductal, beta). This follows the logic that the baseline correlation we are seeing is due largely to zero inflation (i.e., two cells with anywhere from 40% to 80% zero values will show correlation) and not cell-type dependent expression. We acknowledge that using a large PBMC dataset or cell types from various tissues would be interesting, but the PBMC datasets currently available for download all seem to be only from a single individual (inter-individual heterogeneity would not be estimable).

4. What is the rationale behind the distributional choices made in the simulation setup? Specifically, what is the motivation for sampling the final TPM value from a normal distribution (and what is done if the sampled value is negative)?

Previously, we had chosen to use transcripts per million (TPM) values to control for differences in library sizes that would have altered the distributions. We sampled from a normal distribution and recoded negative values to zero. The rationale for this was the large increase in dropout seen for genes with small grand means. We appreciate the reviewer’s point and recognize that the normal distribution is not optimal

given that these data are skewed. In our revised simulation (Fig. 2, Figures S1-S4), we sample from a negative binomial distribution. The dispersion parameters are estimated as a function of the mean, and dropout was modeled as a gamma distribution. We input data that are both raw read counts and per million values. However, we remove correlated genes and control for library size prior to estimating parameters in both situations. Gene-gene correlation was removed to keep large blocks of correlated genes from over-influencing the estimation of the empirical distributions and parameter relationships (e.g., highly-correlated and lowly expressed genes with small grand means would cause a large shift in the maximum likelihood estimation of the gamma distribution for grand means).

5. In general, it is not clear whether the simulated data recapitulates important aspects of real data (and if so, whether it is more representative of data from full-length read count protocols or UMI counts from droplet protocols), including whether the correlation structures shown in Fig. 1 can be recapitulated in the simulated data. A tSNE plot is not sufficient in this respect, since it shows only the relationships among the cells, not the characteristics of the simulated abundances. Showing that the data is representative of real scRNA-seq data is important in order to know whether the conclusions are likely to generalize.

This is a good point. We have included plots (Figures S1-S4) of the many different scRNA-seq data characteristics in real data that we modeled in our simulation. With the exception of the distribution of dropout values, these plots demonstrate how closely the simulated data behave to the real data based on simulation of three different datasets. The correlation is less exaggerated and less irregular (tighter variation) in our simulated data, indicating the effect of intra-individual correlation is likely even larger in real data. We acknowledge our simulated data is not as noisy and does not perfectly capture every aspect of real data. However, this reality is just like any power or type 1 error calculations, and the resulting estimates remain informative.

6. The authors note that DESeq2 failed in many cases because of the normalization method. It is worth noting that DESeq2 implements also another normalization option ('poscounts') which is applicable in situations with sparse count matrices.

We thank the reviewer for pointing out this discrepancy. We have elected to apply DESeq2 in our updated manuscript with the 'pseudo-bulk' approach. This seems to be the most appropriate way to apply their method as DESeq2 was originally designed for bulk RNA-seq data.

7. On p.6, it is written that "A particularly noteworthy approach that has been suggested to account for the within-individual correlation is applying a batch effect correction method", but there is no reference to where this claim was made.

During our review of the single-cell literature, we found that the large majority of papers completed ignored the inter-individual heterogeneity altogether. However, there were a few recent papers that noted differences by donor and attempted to correct for those differences with batch effect correction. The following are examples that use this method in otherwise excellent papers published in quality journals.

- Segerstolpe, Å. et al. Single-Cell Transcriptome Profiling of Human Pancreatic Islets in Health and Type 2 Diabetes. *Cell Metabolism* 24, 593–607 (2016).
- Björklund, Å. K. et al. The heterogeneity of human CD127+ innate lymphoid cells revealed by single-cell RNA sequencing. *Nature Immunology* 17, 451–460 (2016).
- Carmona, S. J. et al. Single-cell transcriptome analysis of fish immune cells provides insight into the evolution of vertebrate immune cell types. *Genome Research* 27, 451–461 (2017).

In an effort not to be abrasive, we purposefully do not explicitly cite these authors in our paper. However, we want to strongly emphasize the importance of not applying batch effect correction for individual prior to DE testing. We note that this is a perfectly reasonable method for removing inter-individual heterogeneity prior to identifying cell types, but we simply wanted to demonstrate and warn authors of using this as a way of correcting for individual effect prior to DE analysis. It is not completely clear whether the batch-effect corrected values were used in downstream DE analysis or not in all of the above papers, but we felt it was still worth addressing since this technique has been commonly applied in areas such as bulk DE testing and differential methylation analysis. In addition, the senior author has seen this method recommended in NIH study sections, a confidential circumstance. We have altered the sentence structure to be more general (pg. 7).

8. How is the performance of the mixed models affected by imbalances in the number of cells between different individuals?

We start by noting that generalized linear mixed models have consistent estimators, and this property does not require balanced data (Jiang, 1998). The question is important, however, as the performance of these models in finite samples merits exploration. Small imbalance in the number of cells between different individuals is captured in our simulation using the Poisson distribution to randomly draw different numbers of cells for each individual. These differences (which grow larger with a larger numbers of cells because the mean and variance for the Poisson distribution are the same) do not affect the performance of the mixed models. The performance of the mixed models will be limited, however, in scenarios where the number of cells grows close to zero for some individuals. In this scenario, we might recommend dropping such individuals from the analysis as they provide very little information to analysis anyways.

9. It's not clear where the results of the extensive literature review (described in the Methods section) are shown.

The goal of the literature review was to get a general idea of how pervasive pseudoreplication was in the current literature. At the time of review, we found very little material about this issue of inter-individual heterogeneity in review papers, found no methods that directly addressed this issue, and found that 100% of the papers implementing the analysis of scRNA-seq data were not properly handling the intra-individual correlation structure. This is generally addressed in the beginning of the paragraph on page 4, but we purposefully did not cite the authors of any of the papers that had been reviewed so as not to be abrasive. The goal of this review was not to directly critique what had been published up to this point, but rather to address a pervasive error that can be readily addressed in future analyses. The list of all reviewed papers can be provided to the reviewers in a separate document if requested.

10. Please provide more detailed information about the data sets used to generate Fig. 1 (only accession numbers are given in the Methods section), such as the technology that was used to generate them, and explain why these data sets were selected.

We have provided more detail about these datasets in the methods section. These datasets were primarily selected because they represented the few publically available datasets that were not cell cultures and had more than two or three individuals. Although more datasets are becoming available now, these datasets illustrate what is intuitively suspected and the point of interest.

Reviewer #2 (Remarks to the Author):

In this paper, Zimmerman and authors examine between cell correlations within individuals in single cell data. The positive correlation observed between cells within an individual is widely accepted in the single cell field, but is usually overlooked in analysis. The authors suggest the use of mixed effects models to improve statistical testing in single cell data.

I think that this paper attempts to address an interesting and often over-looked problem in single cell data analysis. However, I have several concerns about the paper:

1) There is no evidence that the simulation that the authors propose replicates real single cell data. Several simulations have already been proposed for single cell data, as the authors themselves cite in references 12-18, but no comparison has been performed to show that the authors' simulation is "better" than other methods, or more similar to real data. I am also not convinced about the model assumptions and how the TPMs have been generated. I question whether TPMs are indeed normally distributed. Perhaps on a log scale they tend to be normally distributed, but TPMs are highly skewed data.

These are valid points and were raised by Reviewer 1 as was well. Please see our response to item 4) above. Briefly, and as noted there, based on these points we have completely revised the simulation methods and simulation study. We have also documented the similarity to real data.

2) An important reference that is missing is Crowell, H.L., Soneson, C., Germain, P.L., Calini, D., Collin, L., Raposo, C., Malhotra, D. and Robinson, M., 2019. On the discovery of population-specific state transitions from multi-sample multi-condition single-cell RNA sequencing data. *BioRxiv*, p.713412.

- The authors perform an extensive benchmarking study comparing statistical models (including mixed models, as well as MAST) for determining differential gene expression between conditions with multiple individuals in single cell RNA-seq data.**
- They provide software called muscat which provides a simulation platform for multi-sample scRNA-seq data.**

• It is not clear to me what novelty this paper by Zimmerman adds to what has been done in Crowell et al (2019).

We thank the reviewer for noting Crowell et al (2019) in *bioRxiv*. Reviewer 1 also referenced this paper (please also see response to question 2 above). We have included this reference in our paper and have applied their aggregation methods alongside some of the other methods we examined in our paper. We also discuss the costs and benefits of aggregation methods vs mixed models.

We acknowledge this paper does an extremely nice job of:

- Simulating realistic single cell data, particularly relative to having multiple “differential distributions”.
- They compute extensive testing of multiple methods, some of which we elected not to include in our manuscript, such as the Anderson-Darling tests and scDD
- They have focused on identifying “differential states” (i.e., “We refer to these generally as differential state analyses ... to uncover subpopulation-specific changes”)

However, we would like to point out:

- While we acknowledge there is overlap, the very fact that our results coincide and that we have both noticed this strong bias and the problems with treating cells within an individual independently demonstrates the need for new methods. While they propose aggregation, we propose mixed effects models. Each have their own costs and benefits that we now discuss in our manuscript, but both control for a problem that has the potential to tarnish scRNA-seq’s reputation for creating reproducible results.
- There is a standing statistical literature that documents accounting for the correlation within a cluster, here an individual say, over averaging subsamples. A more recent article, Tirrell et al. 2018, explicitly states “that the use the mean of means is either totally inappropriate or less optimal compared to the use of the random effects model” (Snedecor, 1989; Tirrell et al., 2018; Hurlbert, 1984; Millar & Anderson, 2004). ‘Pseudo-bulk’ techniques may control the type 1 error rates and will dramatically decrease compute time. However, we still strongly recommend mixed-effects models based on long-standing statistical justifications for the analysis of subsamples, including increased power. Further, given the costs of the scRNA-seq studies, we are not hitting computational boundaries.
- In Hurlbert (1984) he lays out the main problems with pooling (i.e., sacrificial pseudoreplication):

- “Pooling treatment replicates throws out the information of the variability among replicate plots. Without such information there is no proper way to assess the significant of the difference between treatments.”
- If one carries out a test on the pooled data, one is implicitly redefining the experimental units to be the individual [cells] and not the [individual].”
- “Pooling weights the replicate plots differentially”. In situations of highly imbalanced data, ‘pseudo-bulk’ methods would cause individual cells from individuals with fewer cells to be more heavily weighted than the cells from individuals with more cells unless weighted in some manner (e.g., sample size, variance of the mean). This will be particularly impactful for less common cell types.
- We highlight the following main conclusions from Tirell et al. (2018), pulling from Snedecor & Cochran (1989):
 - For the mean of means (i.e., aggregation methods), “the standard error of the mean (SEM) is the standard deviation of the k means divided by the square root of k. Its expected value is the square root of $\sigma_B^2/k + \sigma^2/kn_h$, where n_h is the harmonic mean of the n_i , i.e. $n_h=k/(1/n_1 + 1/n_2 + \dots, 1/n_k)$.”
 - “For the random effects model, the SEM is the square root of the ANOVA mean square value between subjects divided by the square root of N with an expected value of the square root of $\sigma_B^2/k + \sigma^2/N$.”
 - “For the random effects model, the actual value agrees closely with the target value for all cases. This is because the variance components were calculated explicitly using the random effects model.”
 - “The standard error for the random effects model always estimates a smaller quantity ($\sqrt{\sigma_B^2/k + \sigma^2/N}$) than the standard error of the mean of means ($\sqrt{\sigma_B^2/k + \sigma^2/kn_h}$) since N is always larger than kn_h .”
 - “Using a method other than the random effects model will thus ... over-estimate the standard error and ... lead to an under-commitment of a Type I error, an overestimation of a Type II error and an underestimation of power.”
- In Millar & Anderson (2004), they emphasize the use of mixed-effects models and state that aggregating data “sacrifices statistical power (i.e. inflates the Type II error rate) because it fails to recognize grouping structure in the data.”
- In some ways, this paper nullifies the importance of scRNA-seq and instead implies that cell-sorting followed by bulk RNAseq would be an improvement (in cost and in results). We would like to argue that scRNA-seq still provides great benefit.

- While our simulation does not simulate multiple “differential distributions”, it allows for a more direct example of why treating cells independently can cause problems – particularly after batch effect correction. Our simulated data show the same characteristics of real data (Figures S1-S4).
- Our manuscript provides power calculations – which we emphasize as extremely important. Currently, researchers are building power curves based on the premise that they need to simply reach a baseline number of cells rather than considering how many independent experimental units (i.e., samples) they need. Providing researchers with estimates of power when a hierarchical structure exists remains novel and important. These analyses will, in time, evolve into a tool that researchers can use to compute power analyses for a larger variety of situations.

3) Recently in Crowell et al (2019) as well as in Lun and Marioni (2017) the use of “pseudobulk” samples have been proposed to improve variance estimates. This entails summing the counts of the cells in each cell type per individual and performing statistical tests for differential expression between groups of interest, using already established models such as negative binomial generalised linear models (e.g. using DESeq2 or edgeR), or using normal based models on log-counts such as limma (Ritchie et al, 2015). Performing analysis on pseudobulk samples has the added benefit of substantially reducing the datasets making them quicker and easier to analyse. The authors have not mentioned this strategy, and it should be considered as it is much more computationally tractable for larger single cell datasets, which are becoming increasingly common. I must pose the question, that as datasets continue to get larger with hundreds of thousands of cells, will fitting mixed models be practical?

As noted above, we appreciate the point and adding this component has strengthened our manuscript. We have included ‘pseudo-bulk’ techniques in our manuscript to compare their performance with mixed models. We acknowledge that mixed models will be more computationally intensive but computing capabilities are not a limiting factor here as documented by the large number of simulations. As we emphasized in response to Reviewer 1’s similar point in question 2 above, the mixed model provides more appropriate inference, greater information, and is motivated by robust statistical theory (Snedecor, 1989; Tirrell et al., 2018; Hurlbert, 1984; Millar & Anderson, 2004). These are valuable benefits. We note that if the most appropriate approach to analyzing clustered data and subsampling designs was to aggregate, then there would not have been a need to develop mixed models or generalized estimating equations, methods that are broadly used. Finally, we note that as the number of individuals and cells being analyzed grows, the statistical properties and benefits of mixed models would be further improved.

4) There is no detail about the dataset(s) in the main paper in Figure 1. Is this the only dataset that was analyzed for the purpose of showing strong intra-individual correlation? Are the data generated from droplet-based technology, or plate based? Are the datasets generated on different platforms? There is no other visualization of this data, such as tSNE or UMAP plots, making it difficult to understand the underlying relationships between the cells. Are cells within a cell type more correlated between individuals, than cells within an individual but between cell types? In a complex tissue such as heart or kidney, the correlation structure could be quite different compared to a more homogenous population of cells. There is simply not enough data shown to make broad conclusions about the importance of intra- and inter-individual correlations between cells, and the interplay between tissue complexity, cell types and individuals.

Additional information about the datasets is now explicitly provided beyond just the accession numbers and citation information. All data here is plate based (Fluidigm C1) and tSNE plots have now been included (Figures S1-S4). As the reviewer points out, we also expect this correlation structure to change depending on different cell types. This is why our simulations allow estimating that correlation structure from real data prior to performing downstream analysis or power calculations. As more data become available, it will be interesting to examine how this correlation structure changes across different cell types. These datasets were primarily selected because they represented the few publically available datasets that were not cell cultures and had more than two or three individuals. Although more datasets are becoming available now, these datasets illustrate what is intuitively suspected and the point of interest. Additionally, if the within person correlation is unimportant, then ‘pseudo-bulk’ methods would be wholly inappropriate as that would greatly under-estimate the effective sample size of the study. The fact that we do not *a priori* know for an experiment the impact of the correlations actually argues that the most appropriate method is to account for that correlation until show to be negligible, which the mixed model also handles. Thus, we agree with the idea expressed by the reviewer that there is not enough data available to make a broad conclusion, and we believe this actually argues for the more statistically robust approach of explicitly accounting for the correlation. We acknowledge the value of future exploration of more datasets of larger sample size and varied tissue types (pg. 14).

Minor comments:

1) The paper lacks a coherent structure. This makes the paper more difficult to read, as well as assess the novelty of the proposed solution by the authors. For example, having sections such as simulation study, proposed statistical model, comparison with other methods in the field would be helpful. A lot of detail is in the supplementary and I would prefer more detail in the paper.

We agree with this and we have attempted to address this by adding in section headings, restructuring the whole manuscript, and by adding relevant detail in the manuscript where we felt places would benefit from it. Originally, before being transferred to *Nature Communications*, section headings were discouraged as part of the formatting for this type of manuscript (Letter).

2) The authors don't state the null hypothesis that is being tested, and often refer to "differential expression", or "finding differentially expressed genes". I gather that the authors are referring to a very specific scenario of testing for differentially expressed genes between groups, such as treatment/control, after clustering and cell type identification has been performed. There is no detail regarding whether the testing is performed within cell types. There is also no discussion about the effect of intra-individual correlation on finding marker genes between clusters, which is another often ignored problem in the field.

We have clarified our hypothesis of testing for differentially expressed genes between groups within cell types (pg. 2 & 14). We have added a sentence noting this around the effect of intra-individual correlation on finding marker genes between clusters (pg. 14), but don't want to go too far down that tangent in this manuscript. We completely agree this is a problem, and have hopes to that this problem is also addressed soon.

3) In the Supplementary methods the authors state: "DESeq2 requires integers and at least one gene without a zero value to compute its normalization, so as the number of samples and cells increased, the likelihood of if computing greatly decreased." I would like to point out that edgeR (McCarthy, Chen and Smyth 2012) is another package designed for bulk RNA-seq, that CAN handle TPM/RPKM and doesn't have the same issue as DESeq2's normalization.

We thank the reviewer for pointing out this discrepancy – this is good to know. We have elected to apply DESeq2 in our updated manuscript with the 'pseudo-bulk' approach as this seems to be the most appropriate way to apply this method which was originally designed to work with bulk RNA-seq data.

Citations

Jiang, Jiming. Consistent Estimators in Generalized Linear Mixed Models, *Journal of the American Statistical Association*, 93:442, 720-729 (1998). doi:10.1080/01621459.1998.10473724

Snedecor GW, Cochran WG. *Statistical methods*. Ames, Iowa: Iowa State Univ Press; 1989.

Tirrell, Timothy F et al. “Analysis of hierarchical biomechanical data structures using mixed-effects models.” *Journal of biomechanics* vol. 69, 34-39 (2018). doi:10.1016/j.jbiomech.2018.01.013

Hurlbert, S. H. Pseudoreplication and the Design of Ecological Field Experiments. *Ecological Monographs* 54, 187–211 (1984).

Millar, R. B. & Anderson, M. J. Remedies for pseudoreplication. *Fisheries Research* 70, 397–407 (2004).

REVIEWER COMMENTS

Reviewer #1 (Remarks to the Author):

I think the revised manuscript is substantially improved, and I would like to thank the authors for addressing my concerns. I have a couple of follow-up comments, mostly related to parts where I think a more extensive explanation or description of what was done is warranted.

1. Related to Figure 1:

- How were TPMs estimated? The public records for the utilized data sets don't provide TPMs (only counts and RPKMs). The text also mentions "normalized read count values" - how were these obtained?

- The high correlation in Figure 1 (also between different cell types) is now attributed largely to "zero-inflation" ("two cells with anywhere from 40% to 80% zero values will show correlation"). According to the text all genes with average TPM < 5 were filtered out for Figure 1. How large is the "zero-inflation" based on only the remaining genes?

- Could you clarify whether different individuals/samples in this figure are all from the same condition?

2. Related to the batch correction:

- How was the batch effect correction performed?

- Could you elaborate on the statement that batch effect correction should be used more often "prior to finding marker genes between cell-type clusters"? It would remove inter-individual differences without adjusting the degrees of freedom also in this application.

3. Related to the simulation and application of DE methods:

- How many genes were simulated to be DE?

- Were all genes upregulated in the same group? This can be problematic for methods applying a global normalization assuming that most genes don't change, or at least that some genes are upregulated and others downregulated.

- What type of abundance values (raw counts, TPMs, normalized counts) were used as input to each DE method?

- (related to the previous point) Are all DE methods normalizing the data in the same way after the fold change multiplication? Or do some methods consider the provided data (after multiplying with the fold change) already normalized? This is important for the power comparisons.

- Is the dropout in the simulation independent of the grand mean?

4. All the analyses are done with plate-based data. This should be explicitly mentioned, together with a discussion of whether the conclusions may (or may not) extend to 3' droplet data (with UMIs, lower library sizes and less evident zero-inflation).

5. I have to admit that I still found it confusing to read the Methods section on the literature review, ending with "Each of the methods and implementation articles was thoroughly reviewed and tabled along with its number of citations, date of publication, and any other pertinent information such as number of independent samples, tools used, or number of cells captured.", and not explicitly seeing anywhere in the paper how this information was used. After all, a literature review to find related material and the state of the field is arguably standard practice for any paper. I would suggest to just mention somewhere how the information was used ("From the literature review we concluded that many papers use XXX..." or something along those lines), to make it easier for the reader.

6. The GitHub repository kdzimmer/PseudoreplicationPaper mentioned in the paper does not exist.

Reviewer #2 (Remarks to the Author):

In this revision, the authors have made substantial improvements to their paper, particularly implementing a more appropriate simulation and including aggregation methods in their comparisons. While I agree with the authors that the use of mixed models in order to control for the relationships between cells is sound, it is not backed up by modern data and so has not been evaluated effectively. My main concern about the paper as it stands is that the authors are making very strong recommendations based only on plate-based data, which is known to have different statistical properties to droplet-based datasets, which are far more common in the literature. In its current form the authors would need to make it very clear that the conclusions being drawn are for plate-based single cell data.

1) Simulations: There are at least 10 different single cell simulations available as software. It would be helpful if the authors articulated how their simulation differs to what is already available, and whether it is an improvement. It appears to me that this simulation is very specific to plate-based data.

2) Datasets: The datasets presented are plate-based with relatively few cells represented per individual and not representative of modern data (datasets published in 2016 and 2017). These types of data tend to be more deeply sequenced as well. It is generally accepted that not all single cell data is zero-inflated, so the strongly worded recommendation in this paper for a model that includes zero-inflation may be misleading and inappropriate for droplet-based datasets [Svensson, 2020]. If the authors could include droplet-based datasets and make recommendations for specific mixture models for this type of data, this would greatly strengthen the paper.

a. Svensson, V. Droplet scRNA-seq is not zero-inflated. *Nat Biotechnol* 38, 147–150 (2020).
<https://doi.org/10.1038/s41587-019-0379-5>

3) Power to detect true positives: I agree that this is useful for researchers, but I also wanted to note that there is software that already exists for single cell:

a. Vieth B, Ziegenhain C, Parekh S, Enard W, Hellmann I. powsimR: power analysis for bulk and single cell RNA-seq experiments. *Bioinformatics*. 2017; 33(21):3486-3488.
doi:10.1093/bioinformatics/btx435

4) A missing piece of information is the ranking of the genes from each method along with the false discovery rate. It would be interesting to see whether the gene rankings of the different methods are similar in terms of their false discoveries even though their p-values are not valid when cells within individuals within cell type are highly correlated.

5) I appreciate the discussion regarding computational time in the Discussion section. It would be good to include how long it takes for each method to run on one of the datasets, perhaps the largest one, using `system.time()` in R or similar.

Minor comments:

1) In the literature review, how many datasets were plate-based and how many were droplet-based?

2) Under Simulation in the Methods section, how were the read count values normalised?

3) Could the authors provide a short sentence motivating the choice of a Tweedie distribution?

4) Supplementary Figure 1-4: the axes are too small to read on panels B-G. For panel C, if the y-axis is $\sqrt{\text{dispersion}}$ the relationship may be easier to see.

5) Power curves in the supplementary material: can these be combined into panels in a meaningful way to make it easier for the reader to get the point the authors are making?

6) I note in the tables that the type I error rate for the hurdle mixed model is generally best controlled when the numbers of individuals is at least 10; and that the Tweedie glmm seems to require even larger sample sizes to correctly control the type I error rate. This would be useful to point out in the text. Are there any scenarios where these models fail to compute? i.e. is there a minimum sample size?

Reviewer #1 (Remarks to the Author):

I think the revised manuscript is substantially improved, and I would like to thank the authors for addressing my concerns. I have a couple of follow-up comments, mostly related to parts where I think a more extensive explanation or description of what was done is warranted.

1. Related to Figure 1:

- How were TPMs estimated? The public records for the utilized data sets don't provide TPMs (only counts and RPKMs). The text also mentions "normalized read count values" - how were these obtained?

The data downloaded were already processed and estimated as TPMs. For example, after downloading and extracting the following <https://www.ebi.ac.uk/arrayexpress/experiments/E-MTAB-5061/E-MTAB-5061.processed.1.zip>, there is a file labeled "E-MTAB-5061.expression_tpm.mtx" which contains the TPM values for each gene and each cell. In the text, normalized read count values refers to the TPMs. We understand this is not a true normalization, so to avoid confusion, we have simply changed "normalized read count values" to "TPMs" (pg. 5, figure 2; pg. 16, line 336; pg. 18, lines 367 & 371).

- The high correlation in Figure 1 (also between different cell types) is now attributed largely to "zero-inflation" ("two cells with anywhere from 40% to 80% zero values will show correlation"). According to the text all genes with average TPM < 5 were filtered out for Figure 1. How large is the "zero-inflation" based on only the remaining genes?

We apologize for not updating this section of the text to accurately reflect the updates made to the analysis. We have adjusted the text so that it now more accurately reflects the new analysis after the first round of review. Prior to removing correlated genes, we were filtering genes with an average TPM < 5. Now we are only removing genes with an average TPM = 0, and then removing the correlated genes (pg. 16, line 337).

- Could you clarify whether different individuals/samples in this figure are all from the same condition?

Each of the datasets used in this manuscript represent individuals from varying conditions. For the colorectal cancer dataset, all tissue was taken from healthy mucosa. The B-cells and T-cell datasets sampled tumor tissue from 11 and 17 individuals, respectively, all with melanoma. The pancreatic cells were sampled from 10 individuals, 6 of which are healthy controls and 4 are patients with type 2 diabetes. We have clarified this in the text of the methods section where we briefly describe each dataset (pg. 17, lines 352-358).

2. Related to the batch correction:

- How was the batch effect correction performed?

We used ComBat and adjusted for individual as batch. We cited ComBat in our manuscript, but we now more explicitly provide this information in our manuscript (pg. 19, line 398).

- Could you elaborate on the statement that batch effect correction should be used more often "prior to finding marker genes between cell-type clusters"? It would remove inter-individual differences without adjusting the degrees of freedom also in this application.

We apologize for the lack of clarity, and we have revised the text (pg. 10, line 207). To clarify, we advise removing inter-individual differences prior to cell-type clustering and then subsequently conducting differential expression analysis on the cell-type clusters with mixed models to account for intra-individual correlation on the original/uncorrected data to account for inter-individual differences.

3. Related to the simulation and application of DE methods:

- How many genes were simulated to be DE? Were all genes upregulated in the same group? This can be problematic for methods applying a global normalization assuming that most genes don't change, or at least that some genes are upregulated and others downregulated.

We simulated all genes as DE for each set of runs. Originally, we did not anticipate this would be a problem because we simulated genes to either be upregulated or downregulated at equal frequency. In actuality, however, this was not the case. We thank the reviewer for observing this error, and we recognize this as a large problem – particularly after we observed the large differences in DESeq2's size factors between cases and controls. We have updated all of our 'pseudo-bulk' runs to correct for this by fixing all of the size factors to one. As a result, we include updated and expanded power calculations for pseudo-bulk methods (Figures S5-S7, pg. 4, line 69; pg.8, line 160; pg. 11, lines 216-246). We deeply appreciate this reviewer's insight.

- What type of abundance values (raw counts, TPMs, normalized counts) were used as input to each DE method?

Most methods used a $\log(x + 1)$ transform of the raw counts, which is what was also used with the tweedie mixed model. For DESeq2, the built-in normalization was used. We have added a sentence in the methods section to more explicitly state the approaches (pg. 19, line 404). As pointed out in the previous comment, this is problematic and we have now adjusted our approach (pg. 20, line 429).

- (related to the previous point) Are all DE methods normalizing the data in the same way after the fold change multiplication? Or do some methods consider the provided data (after multiplying with the fold change) already normalized? This is important for the power comparisons.

All methods, except for the "pseudo-bulk" methods, were being applied to the simulated counts after a $\log(x + 1)$ transformation (without normalization after the fold change

multiplication). Normalization is not required in these data because a difference in library size was not simulated here. DESeq2's normalization is the reason the power calculations were so different between the pseudo-bulk and mixed models. Once correcting for the normalization differences by fixing all of the size factors to one, this problem was resolved, although 'pseudo-bulk' techniques are still not as well powered as mixed effects models when there is imbalance. As noted above, we include updated and expanded power calculations for the pseudo-bulk and mixed model comparisons (Figures S5-S7, pg. 4, line 69; pg.8, line 160; pg. 11, lines 216-246).

- Is the dropout in the simulation independent of the grand mean?

Yes, there are two types of zeros that occur in our simulation. The first set of zeros are due to a small grand mean for the negative binomial distribution. These would be zeros due to lowly expressed genes. The second set of zeros (the dropout that is simulated independently in this case) corresponds to genes "missing completely at random" due to technological error or stochastic expression. We completed extensive investigations to enable our simulation engine to best approximate real data. We understand this may be the case for plate-based data only and that a recent publication has determined that technical dropout does not exist in droplet data (Svensson, 2020). However, our results translate to droplet data as the hierarchical correlation structure will still exist, just without the independent zero-inflation adding noise. The use of mixed models for clustered data without a zero inflation component is common practice in statistics.

4. All the analyses are done with plate-based data. This should be explicitly mentioned, together with a discussion of whether the conclusions may (or may not) extend to 3' droplet data (with UMIs, lower library sizes and less evident zero-inflation).

We now explicitly mention that the simulations were based on patterns observed with plate-based data but that droplet data will also exhibit within-person correlations and the methods are applicable to these data (pg. 16, line 309). Specifically, single-cell data will always have a hierarchical correlation structure due to the way these data are collected. In fact, the reduction in technical noise with these newer technologies should only serve to further elucidate the hierarchical correlation structure in these data. In the future, as more droplet based data become publically available, our simulation will be adjusted to mimic droplet data, particularly the zero-inflation component.

5. I have to admit that I still found it confusing to read the Methods section on the literature review, ending with "Each of the methods and implementation articles was thoroughly reviewed and tabled along with its number of citations, date of publication, and any other pertinent information such as number of independent samples, tools used, or number of cells captured.", and not explicitly seeing anywhere in the paper how this information was used. After all, a literature review to find related material and the state of the field is arguably standard practice for any paper. I would suggest to just mention somewhere how the information was used ("From the literature review we concluded that many papers use XXX..." or something along those lines), to make it easier for the reader.

We have modified the sentence in the introduction to be more explicit about how we used to the literature review to reach the conclusion that pseudoreplication was, at the time of review, nearly ubiquitous in single-cell DE analysis (pg 2, line 49).

6. The GitHub repository `kdzimmer/PseudoreplicationPaper` mentioned in the paper does not exist.

We apologize for the error. This should be “`kdzimm/`” not “`kdzimmer/`”.

Reviewer #2 (Remarks to the Author):

In this revision, the authors have made substantial improvements to their paper, particularly implementing a more appropriate simulation and including aggregation methods in their comparisons. While I agree with the authors that the use of mixed models in order to control for the relationships between cells is sound, it is not backed up by modern data and so has not been evaluated effectively. My main concern about the paper as it stands is that the authors are making very strong recommendations based only on plate-based data, which is known to have different statistical properties to droplet-based datasets, which are far more common in the literature. In its current form the authors would need to make it very clear that the conclusions being drawn are for plate-based single cell data.

1) Simulations: There are at least 10 different single cell simulations available as software. It would be helpful if the authors articulated how their simulation differs to what is already available, and whether it is an improvement. It appears to me that this simulation is very specific to plate-based data.

There are a number of single-cell simulations available as software, but none of them are modeling the inter-individual differences from real single-cell data. Splatter, the most cited single-cell simulation tool, shows great similarity to real data and great improvements upon other simulation tools, but never evaluates how closely it models the hierarchical correlation structure of single-cell data. Their tool allows for the simulation of batch effects, which could potentially be used to introduce a hierarchical correlation structure where batches are individuals, but in that scenario, the correlation structure would not be explicitly modeled after real data the way ours is. The simulation tool referenced in the previous round of reviews, muscat, is the only other simulation tool that explicitly models the inter-individual differences (Crowell et al – biorxiv). We note that, as of writing this, their work is still only published on a pre-print server. This doesn't diminish the value of their work, or the relevance of their simulation tool, but rather presents two independent sets of researchers who recognized an important need in the field and are working to resolve it.

2) Datasets: The datasets presented are plate-based with relatively few cells represented per individual and not representative of modern data (datasets published in 2016 and 2017). These types of data tend to be more deeply sequenced as well. It is generally accepted that not all single cell data is zero-inflated, so the strongly worded recommendation in this paper for a model that includes zero-inflation may be misleading and inappropriate for droplet-based datasets [Svensson, 2020]. If the authors could include droplet-based datasets and make recommendations for specific mixture models for this type of data, this would greatly strengthen the paper.

The major point of our manuscript is to state the importance of using random effects models to account for the hierarchical structure that will exist in all single-cell RNA-seq data – not just plate-based data. In this manuscript, the two-part hurdle model is our recommendation as a means of accounting for the sparsity of the data, but it is the use of random effects to account for inter-individual heterogeneity that we strongly emphasize in this manuscript.

That said, we do believe that droplet-based data is still sparse. Svensson's manuscript states that the sparsity is no longer due to technical errors. The fact that droplet-based technology has very few "technical zeros" is actually all the more reason for one to apply a two-part hurdle model. In this scenario, where the technical noise is not present, the zeros are actually true zeros, so testing for differences in the proportions of zeros across cells is actually more meaningful. Identifying genes that differ in the proportion of on/off genes as well as in magnitude of expression is a great advantage of applying a two-part hurdle model that models the zero-inflation due to biological variation and not technical molecular inefficiencies. In the scenario that very few zeros, or even no zeros are observed, the continuous component of the two part-hurdle model will still compute. We also note that a large number of mixed models with other distributional assumptions exist. So, while a two-part hurdle model is applicable to the data observed herein, if the data for specific cell-types or other single-cell technologies behave differently than what is observed here, the distributional assumptions for the mixed model should also change.

**3) Power to detect true positives: I agree that this is useful for researchers, but I also wanted to note that there is software that already exists for single cell:
a. Vieth B, Ziegenhain C, Parekh S, Enard W, Hellmann I. powsimR: power analysis for bulk and single cell RNA-seq experiments. *Bioinformatics*. 2017;33(21):3486-3488. doi:10.1093/bioinformatics/btx435**

The program powsimR simulates cells independently and makes power recommendations on the number of cells (they use the term 'replicates') a researcher should obtain to reach a certain power rather than the number of independent experimental units (individuals). This is the critical difference between the power curves represented in our manuscript and the power calculations that powsimR provides. We also acknowledge another available single-cell RNA-seq power calculator, SCOPEIT. This tool also provides recommendations for the number of total cells required before the researcher obtains a minimum number of a rare cell type of interest. This is a separate question than the one we are trying to inform with our power calculations.

4) A missing piece of information is the ranking of the genes from each method along with the false discovery rate. It would be interesting to see whether the gene rankings of the different methods are similar in terms of their false discoveries even though their p-values are not valid when cells within individuals within cell type are highly correlated.

This is a thoughtful suggestion best served by a more thorough investigation in subsequent papers. However, we did briefly explore if the rankings of 2000 genes were preserved across the different methods evaluated in this manuscript. With stochastically balanced data, the rank-order correlation coefficients across methods are:

Spearman's rank-order correlation		Pseudobulk		Two-part hurdle			Tweedie				
		Mean	Sum	Default	Corrected	GLMM	GLM	GLMM	GEE1	Tobit	ROTS
Pseudobulk	Mean	1.00	0.99	0.83	0.25	0.96	0.58	0.85	0.85	0.58	0.73
	Sum	0.99	1.00	0.81	0.22	0.95	0.54	0.82	0.82	0.54	0.71
Two-part hurdle	Default	0.83	0.81	1.00	0.13	0.85	0.81	0.79	0.86	0.67	0.77
	Corrected	0.25	0.22	0.13	1.00	0.27	0.25	0.37	0.36	0.24	0.19
	GLMM	0.96	0.95	0.85	0.27	1.00	0.59	0.83	0.84	0.60	0.69
Tweedie	GLM	0.58	0.54	0.81	0.25	0.59	1.00	0.75	0.85	0.65	0.73
	GLMM	0.85	0.82	0.79	0.37	0.83	0.75	1.00	0.96	0.60	0.69
	GEE1	0.85	0.82	0.86	0.36	0.84	0.85	0.96	1.00	0.67	0.73
	Tobit	0.58	0.54	0.67	0.24	0.60	0.65	0.60	0.67	1.00	0.61
	ROTS	0.73	0.71	0.77	0.19	0.69	0.73	0.69	0.73	0.61	1.00

There is strong positive correlation among the two pseudo-bulk methods and decent correlation among the GLMM and GEE1 methods. However, it appears that ranks are not well-preserved across methods and that identifying the top statistical associations for each method will not reveal identical results. Again, this is simulated stochastically balance. As the imbalance in the number of cells per person increases, the methods should show even greater departure.

We note that, even if rankings were preserved, the false discovery rate assumes an unbiased distribution of p-values and will be inflated if there is a bias. In this scenario, the false discovery rate for the methods analyzing cells independently will be heavily biased.

5) I appreciate the discussion regarding computational time in the Discussion section. It would be good to include how long it takes for each method to run on one of the datasets, perhaps the largest one, using `system.time()` in R or similar.

We have added a table to the supplemental material (Table S5) that includes how long each method takes to analyze various sizes of data. We note that if one is concerned about these run times for the generalized linear mixed effects models, that these times are all based on R run times and they are not parallelized. Python and SAS are examples of other tools that are able to compute generalized linear mixed effects models with greater speed and computational power if necessary.

Minor comments:

1) In the literature review, how many datasets were plate-based and how many were droplet-based?

Unfortunately, we did not tabulate this information at the time of our literature review. Regardless of whether or not data are plate-based or droplet-based, scRNA-seq data will have a hierarchical correlation structure where cells within an individual are more highly correlated than cells across individuals. Therefore, the manuscripts we reviewed that did not account for the within-sample correlation in tests of DE, regardless of data type, should not have been analyzing cells independently.

2) Under Simulation in the Methods section, how were the read count values normalised?

In the text, normalized read count values refers to the TPMs. We understand this is not a true normalization, so to avoid confusion, we have simply changed “normalized read count values” to “TPMs” (pg. 5, figure 2; pg. 16, line 336; pg. 18, lines 367 & 371).

3) Could the authors provide a short sentence motivating the choice of a Tweedie distribution?

We have inserted a small explanation into the manuscript describing our reasoning behind using a Tweedie distribution (pg. 13, line 272).

4) Supplementary Figure 1-4: the axes are too small to read on panels B-G. For panel C, if the y-axis is $\sqrt{\text{dispersion}}$ the relationship may be easier to see.

This is appreciated, thank you. We have increased the resolution of these figures, put the figures in the landscape orientation to increase the size, and have provided even more detail in the figure legend regarding these panels. We have fixed the y-axis in panel C as recommended. We note that the axes for each set of panels across the real and simulated data are held constant and are primarily meant to demonstrate the similarity between the real and simulated data.

5) Power curves in the supplementary material: can these be combined into panels in a meaningful way to make it easier for the reader to get the point the authors are making?

These power curves are meant to provide other researchers with a rapid way to look up and approximate their power given a certain sample size and number of cells per individual. While we acknowledge that these chew up a lot of space in the supplementary material, we believe keeping these power curves large and easy to read is in the best interest of readers.

6) I note in the tables that the type I error rate for the hurdle mixed model is generally best controlled when the numbers of individuals is at least 10; and that the Tweedie glmm seems to require even larger sample sizes to correctly control the type I error rate. This would be useful to point out in the text. Are there any scenarios where these models fail to compute? i.e. is there a minimum sample size?

If there is just a single individual per treatment group, the mixed-model will fail. In this case, the generalization to a larger population needs to be made based on non-sampling, non-statistical grounds. More generally, the reviewer is correct that performance of mixed models improves with more than a few independent individuals. With smaller sample sizes and fewer cells, the likelihood of complete separation greatly increases, particularly with the discrete component of the two-part hurdle model. In these scenarios, mixed-models will not compute without small perturbations to the data. For example, with two individuals per treatment group the mixed-model will compute for some genes, but many genes will fail to compute due to complete separation. This is particularly true when there are few (< 50) cells per individual. The proportion of genes that fail rapidly decreases with an increase in independent individuals. We do not provide a recommendation on the minimum number of independent individuals or minimum number of cells per person per se. Rather, our simulations provide guidance to researchers on the performance of the methods. Further, ignoring the within-person correlations

might allow analyses to be completed but the inference would be incorrect (i.e., inflated type 1 error rate). We have added a small part in the discussion about these issues (pg. 14, line 288).

REVIEWER COMMENTS

Reviewer #1 (Remarks to the Author):

I appreciate that the authors have expanded the discussion section in the revised manuscript, contrasting different types of approaches. The authors have also provided crucial clarifications of the simulation procedure. I think, however, that additional clarifications are required for a reader to be able to fully understand what was done. I have a few specific comments:

1. The abundance unit usage needs clarification. The methods section mentions "TPM read count values", but TPMs are not read count values. In addition, the methods section states that a "read count value" was drawn from a NB distribution with an expected value equal to the mean estimated from the provided TPMs. This implies an expected library size of 1 million, which may or may not be realistic for single-cell data (depending on the protocol). Importantly, the absolute count (not just the relative abundance) matters for count-based DE methods, and I wonder how this affects the evaluation.
2. The description of the simulation still needs to be improved - it's not clear whether the genes are always upregulated in the same condition, nor how many genes are DE (the text says that the global mean expression of 'a gene' is modified). It should also be clarified how many genes are simulated in each data set. I would recommend to state explicitly that this is an unusual simulation setting (since all genes are DE), and motivate why it was done like this rather than in a more conventional way, attempting to replicate the observations of an experimental data set. I'm concerned by this since tools like DESeq2 are developed with certain assumptions in mind, and are designed to combine information from a large number of genes when estimating dispersions. Thus, it is unclear to me whether the results obtained here are representative of real data sets.
3. Even though the GitHub repository doesn't contain a formal 'software package', please still assign a license to the provided code so that it is clear how it can be reused.
4. I don't think that statements like "only capture biological zeros of interest as suggested in droplet-based sequencing methods", "droplet-based single-cell methods greatly decrease the amount of technical noise" are correct. The apparent lack of zero-inflation in data from droplet protocols doesn't mean that each zero represents true non-abundance; rather, the sequencing depth is so low that the zeros can be explained by sampling from a regular count distribution with low mean. So many of the zeros are still 'technical' (and would disappear with deeper sampling/sequencing), but do not require an additional zero-inflation component for accurate modeling.
5. The axis titles in the individual panels in Figs. S1-S4 are still very hard to read.

Reviewer #2 (Remarks to the Author):

The authors have made great efforts to incorporate the reviewers' suggestions and as such the revised article is much improved. I still feel that the article would be more complete by including datasets generated from a droplet based technology such as 10x Chromium data, but as this is now clearly stated in the text I am more comfortable with the conclusions of the study.

I have some additional comments:

1. I attempted to view the code at the github repository (kdzimmer/PseudoreplicationPaper), however the repository does not exist or at least was not accessible to me. Hence I was unable to review the code or reproduce the analysis.
[ED: I just tried to visit the Github repository and was able to access it, so I'm not sure what the problem was. Perhaps include a full link in your code availability statement.]
2. The legend in Figure 1 could use an explanation of the random pancreatic cells boxplots and the

meaning with regards to the rest of the figure and link with the text.

3. For Table 1 (and other type I error tables in supplementary material), are there standard deviations associated with the type I error rates? This is not explicitly stated in the table caption.

4. It should be stated in Table 1 that combat was used for batch correction, as well as in the main text on pg 7 line 133. There are many single cell specific batch correction techniques, such as harmony, liger and Seurat integration. Is there a particular reason why the authors chose combat over more recent single cell specific techniques?

5. I'm a bit concerned about the power simulations only including genes that are differentially expressed as this is not representative of real scRNA-seq data. It was not clear from the description in the methods section exactly how this was performed (i.e. number of genes DE, number of genes up-regulated and down-regulated). If genes are included that are not changing between conditions then the sensitivity and specificity of each method can be calculated. I feel that this is an important aspect that is missing from the paper as it currently stands.

6. Supplementary figure S1 looks identical to the previous revision although the authors have said it was modified in their comments to reviewers.

7. Are Table S1 and Table 1 the same?

8. What are the units in Table S5? Are they seconds/minutes?

9. I'm not sure about the statement hypothesizing the correlation increasing due to zero-inflation (pg4 line 89). If additional zeroes are stochastic then this statement is not true.

10. Thank you for the table showing the ranking of the genes between the different methods. I found it quite enlightening. One result that really stood out was that the corrected 2-part hurdle model is clearly showing a very different ranking compared to every other method with correlations around 0.2. Something is clearly going wrong in this analysis. The strongest correlations (>0.95) are between the pseudobulk methods and the two part hurdle model GLMM, and the tweedie GLMM and the GEE1 models. Did the authors include the tweedie GLMM and GEE1 models in the power calculations? How does the correlation change when the methods are compared to the "true" rankings? For example, ranking the genes by their simulated logFC?

11. Did you run muscat to see if it could recapitulate the statistical properties of the real data as well as your simulation? The countsimQC Bioconductor package can be used to compare multiple count matrices (i.e. simulated data) against the real data to see whether the simulated data recapitulates properties of the real data.

Division of Public Health Sciences
Department of Biostatistics and Data Science

October 9th, 2020

Dear Reviewers,

We sincerely thank you for your most recent review of our manuscript, as you have helped to improve its quality and clarity. We also sense an urgency to this research, as there continue to be publications that do not account for the within-person correlation inherent in single-cell data.

In this revision, we have carefully considered your comments and revised our manuscript accordingly. Most importantly, we have increased the clarity of our simulation methods, improved our figures, added a supplemental table, and added the evaluation of some 10x Chromium data.

In addition, we have reworked the text to better emphasize our two major points. Our first, and most important, point is that tests of hypotheses for differential expression must account for the within-person correlation that exists in single-cell RNA-seq data. The second point is that mixed-effects models are currently the best way of accounting for the within-person correlation. In this manuscript, the two-part hurdle model is employed as an easily implementable means of accounting for sparse data, but it is the use of mixed-effects models that is key. If the single-cell data being analyzed do not exhibit zero inflation, then a mixed-effects model can be applied using a more appropriate distribution.

Below we provide a point-by-point response to each of your comments. All of the changes we made in the manuscript are highlighted in yellow.

Again, we greatly appreciate your insights, time, and efforts.

Sincerely,

Carl D. Langefeld, Ph.D.
Professor, Department of Biostatistics and Data
Science
Division of Public Health Sciences
Wake Forest School of Medicine
Medical Center Boulevard, WC 2321
Winston-Salem, NC 27157-1063

Kip D. Zimmerman, B.A.S.
PhD Candidate, Molecular Genetics and
Genomics
Wake Forest School of Medicine
Medical Center Boulevard, WC 2321
Winston-Salem, NC 27157-1063

Reviewer #1 (Remarks to the Author):

I appreciate that the authors have expanded the discussion section in the revised manuscript, contrasting different types of approaches. The authors have also provided crucial clarifications of the simulation procedure. I think, however, that additional clarifications are required for a reader to be able to fully understand what was done. I have a few specific comments:

We thank the reviewer for recognizing our efforts to improve the manuscript. In accord with your review, we have worked to polish our description of the simulation procedure to enable readers to completely understand our approach.

1. The abundance unit usage needs clarification. The methods section mentions "TPM read count values", but TPMs are not read count values. In addition, the methods section states that a "read count value" was drawn from a NB distribution with an expected value equal to the mean estimated from the provided TPMs. This implies an expected library size of 1 million, which may or may not be realistic for single-cell data (depending on the protocol). Importantly, the absolute count (not just the relative abundance) matters for count-based DE methods, and I wonder how this affects the evaluation.

We have modified "TPM read count values" to "TPM values". Your interpretation is correct that a value was drawn from a NB distribution with an expected value equal to the mean estimated from the provided TPMs, which does imply an expected library size of 1 million. We realize this is not necessarily realistic if one were interested in simulating data with varying library sizes. Although an interesting question, it is beyond the intent of this manuscript. Our simulations were not designed to observe the effects of differing library sizes on downstream analysis. Rather, they control for differing library sizes to reduce unwanted noise to enable us to sharply articulate the need to account for within-person correlations when completing tests of differential expression in single-cell data. We have clarified this point in our Methods section (pp. 19-23, lines 388-484).

2. The description of the simulation still needs to be improved - it's not clear whether the genes are always upregulated in the same condition, nor how many genes are DE (the text says that the global mean expression of 'a gene' is modified). It should also be clarified how many genes are simulated in each data set. I would recommend to state explicitly that this is an unusual simulation setting (since all genes are DE), and motivate why it was done like this rather than in a more conventional way, attempting to replicate the observations of an experimental data set. I'm concerned by this since tools like DESeq2 are developed with certain assumptions in mind, and are designed to combine information from a large number of genes when estimating dispersions. Thus, it is unclear to me whether the results obtained here are representative of real data sets.

We appreciate the Reviewer's point and have clarified our simulation procedures. Specifically, we have now divided the Methods section pertaining to the simulation procedures into four parts: "Simulation engine", "Evaluation of type 1 error", "Evaluation of power", and

“Evaluation of rank-order preservation”. These provide a more precise explanation of how the various simulations and evaluations were completed (pp. 18-23, lines 385-500).

The main point of emphasis in this paper is the critical importance of accounting for within-person correlation when testing for associations. As previously discussed, this is well established in classic statistical theory and application. This is a timely and important question as there continue to be publications of single-cell RNA-seq data that do not do so, and therefore their results cannot be replicated as they are type I errors. Simulating the entire transcriptome is not necessary to demonstrate this point. All the methods we tested in this manuscript, except the “pseudo-bulk” methods with DESeq2, compute the tests one gene at a time without integrating information across all genes. This approach is commonly used in other genomic and other –omic (i.e., high-dimensional datasets) calculations of type I error rates and power, including other RNA-seq power calculators such as RnaSeqSampleSize¹ and RNAseqPS². Thus, we are consistent with their approaches.

¹Zhao, S., Li, C.-I., Guo, Y., Sheng, Q. & Shyr, Y. RnaSeqSampleSize: real data based sample size estimation for RNA sequencing. *BMC Bioinformatics* **19**, 191 (2018).

²Guo, Y., Zhao, S., Li, C.-I., Sheng, Q. & Shyr, Y. RNAseqPS: A Web Tool for Estimating Sample Size and Power for RNAseq Experiment. *Cancer Inform* **13**, 1–5 (2014).

3. Even though the GitHub repository doesn't contain a formal 'software package', please still assign a license to the provided code so that it is clear how it can be reused.

An MIT license has been added to GitHub.

4. I don't think that statements like "only capture biological zeros of interest as suggested in droplet-based sequencing methods", "droplet-based single-cell methods greatly decrease the amount of technical noise" are correct. The apparent lack of zero-inflation in data from droplet protocols doesn't mean that each zero represents true non-abundance; rather, the sequencing depth is so low that the zeros can be explained by sampling from a regular count distribution with low mean. So many of the zeros are still 'technical' (and would disappear with deeper sampling/sequencing), but do not require an additional zero-inflation component for accurate modeling.

We greatly appreciate the Reviewer taking the time to clarify this. We understand and agree. We have removed these statements so that we no longer overstate the amount of technical noise that is decreased in droplet-based sequencing data. In this manuscript, we implemented a two-part hurdle model as an immediately implementable means of accounting for sparse data, but it is the use of random effects to account for the within-person correlation that we emphasize. If single-cell data do not exhibit zero inflation, then the mixed model can be applied using a more appropriate distribution. We now explicitly state this point in our manuscript (pp. 14, lines 294-295). We reiterate that the major point of our manuscript is to state the importance of using random effects models or other appropriate statistical methods (e.g., aggregation techniques, GEE1, resampling techniques) to account for the hierarchical structure that will exist in all single-cell RNA-seq data.

5. The axis titles in the individual panels in Figs. S1-S4 are still very hard to read.

We have restructured the figures to enable the axes to be more easily read.

Reviewer #2 (Remarks to the Author):

The authors have made great efforts to incorporate the reviewers' suggestions and as such the revised article is much improved. I still feel that the article would be more complete by including datasets generated from a droplet based technology such as 10x Chromium data, but as this is now clearly stated in the text I am more comfortable with the conclusions of the study.

We appreciate the Reviewer recognizing our efforts to improve the quality and clarity of our manuscript. Given the Reviewer's clear expertise in this area and her/his sense of importance of these data to the question, we have come to agree that this manuscript would benefit from the inclusion of 10x Chromium data. Thus, we have now downloaded 10x Chromium data and document the positive within-person correlation (Figure 1).

I have some additional comments:

1. I attempted to view the code at the github repository (kdzimmer/PseudoreplicationPaper), however the repository does not exist or at least was not accessible to me. Hence I was unable to review the code or reproduce the analysis. [ED: I just tried to visit the Github repository and was able to access it, so I'm not sure what the problem was. Perhaps include a full link in your code availability statement.]

Following the suggestion of the editor, a full link is now provided in the manuscript. For ease of access, the link to the repository is: <https://github.com/kdzimm/PseudoreplicationPaper>.

2. The legend in Figure 1 could use an explanation of the random pancreatic cells boxplots and the meaning with regards to the rest of the figure and link with the text.

These boxplots are a random sampling of the other pancreatic cell types included in Figure 1, rather than only partitioning by one cell type. They were included to satisfy a reviewer's previous comments. We now explain "random cell types" in the figure legend (pp. 4) in addition to the Methods section (pp. 17-18, lines 363-367). This figure now includes additional cell types, based on our 10x Chromium data.

3. For Table 1 (and other type I error tables in supplementary material), are there standard deviations associated with the type I error rates? This is not explicitly stated in the table caption.

We added 95% confidence bounds to the type 1 error rate in Supplemental Table S1. These confidence bounds are computed based on a sample size of 250,000 and are quite narrow. As noted in the table, the 95% confidence bounds are available in the supplementary material.

4. It should be stated in Table 1 that combat was used for batch correction, as well as in the main text on pg 7 line 133. There are many single cell specific batch correction techniques, such as harmony, liger and Seurat integration. Is there a particular reason why the authors chose combat over more recent single cell specific techniques?

We now directly state that we used ComBat (Table 1; pp. 7 line 135; pp. 11 line 225). The Reviewer is correct, there are numerous single-cell specific batch correction techniques. We chose it because it was used in some of the single-cell papers we had reviewed. All of these methods play a very important role in removing batch-effects prior to data integration. However, the underlying concept for all of them is that they should not be applied as a means of accounting for within-person correlation when estimating the variance used in testing the hypothesis of differential expression. We now explicitly state this point in our manuscript (pp. 11-12, lines 225-228).

5. I'm a bit concerned about the power simulations only including genes that are differentially expressed as this is not representative of real scRNA-seq data. It was not clear from the description in the methods section exactly how this was performed (i.e. number of genes DE, number of genes up-regulated and down-regulated). If genes are included that are not changing between conditions then the sensitivity and specificity of each method can be calculated. I feel that this is an important aspect that is missing from the paper as it currently stands.

We have added more explanation within the “Simulation” section of our methods (please see response to comment #2, Reviewer 1). We also edited our methods section to be more precise about how the simulations were completed (pp. 18-23, lines 385-500).

In our simulation, each gene is simulated independently and, subsequently, the test (with the exception of DESeq2) for each gene is computed independently. Therefore, power can be estimated by repeatedly simulating a gene for a given fold-change, until a stable approximation of power at that fold-change is reached. This approach is commonly used in other genomic and other –omic (i.e., high-dimensional datasets) power calculations, including other RNA-seq power calculators such as RnaSeqSampleSize¹ and RNAseqPS². Thus, we are consistent with their approaches for estimating power.

In the previous review, we had simulated a dataset that included some genes under the alternative hypothesis with varying degrees of fold-change and some genes that were under the null hypothesis to compute the rank-order correlation between the results of the various methods applied in this manuscript (reviewer’s comment 10 below). We have now expanded on this simulation (see Methods). Here, 2,000 independent genes were simulated and independently analyzed for each incremental increase in fold-change between 0 and 4. As the reviewer noted, this dataset also provided an opportunity for computing sensitivity. Specificity can be computed as $1 - \text{type 1 error}$, so when the type 1 error rate for a method is inflated, the specificity of the method is small. We computed sensitivity for the two-part hurdle GLM, the two-part hurdle GLMM, and the two-part hurdle GLM applied after batch-effect correction to account for within-person correlation. These three methods were selected because they provide an ideal contrast of the three ways of handling the within-person correlation while holding constant the underlying

distributional assumptions of the model. The GLM ignores the within-person correlation, the GLMM properly accounts for it, and the GLM with batch-effect correction improperly accounts for it. As expected, when within-person correlation is not accounted for, enormous type 1 error rates cause sensitivity to appear very high. Without considering the specificity (1 - type 1 error), one might think such methods are an improvement due to higher sensitivity at the lesser fold-changes (Fig. S5). However, the specificity (1 - type 1 error) for these methods is extremely low because of the large number of false-positive results. We now include these results in the manuscript (pp. 7-8, lines 138-149; pp. 11, lines 212-215).

¹Zhao, S., Li, C.-I., Guo, Y., Sheng, Q. & Shyr, Y. RnaSeqSampleSize: real data based sample size estimation for RNA sequencing. *BMC Bioinformatics* **19**, 191 (2018).

²Guo, Y., Zhao, S., Li, C.-I., Sheng, Q. & Shyr, Y. RNAseqPS: A Web Tool for Estimating Sample Size and Power for RNAseq Experiment. *Cancer Inform* **13**, 1-5 (2014).

6. Supplementary figure S1 looks identical to the previous revision although the authors have said it was modified in their comments to reviewers.

We have completely reconstructed Supplemental Figures S1-S4.

7. Are Table S1 and Table 1 the same?

Yes, they were previously identical. However, based on the Reviewer's comment above regarding standard deviations associated with the type 1 error rates, we have now added confidence bounds to Supplemental Table S1.

8. What are the units in Table S5? Are they seconds/minutes?

We have clarified in the Supplemental Table S5 legends that the units are seconds.

9. I'm not sure about the statement hypothesizing the correlation increasing due to zero-inflation (pg4 line 89). If additional zeroes are stochastic then this statement is not true.

We have removed this statement.

10. Thank you for the table showing the ranking of the genes between the different methods. I found it quite enlightening. One result that really stood out was that the corrected 2-part hurdle model is clearly showing a very different ranking compared to every other method with correlations around 0.2. Something is clearly going wrong in this analysis. The strongest correlations (>0.95) are between the pseudobulk methods and the two-part hurdle model GLMM, and the tweedie GLMM and the GEE1 models. Did the authors include the tweedie GLMM and GEE1 models in the power calculations? How does the correlation change when the methods are compared to the "true" rankings? For example, ranking the genes by their simulated logFC?

We anticipated that the strongest correlations would be observed among methods that appropriately account for the within-person correlation. This set includes the pseudo-bulk methods, which are just slightly less efficient. The reviewer's point enables us to underscore the

importance of accounting for the within-person correlation again. If this is not done, the analysis results in inflated p-values and spurious results that do not maintain rank order – particularly if a batch effect correction is used to correct for the within person correlation. We have now completed a more thorough investigation of the gene rankings (see Table S5). We reference these data in our Results (pp. 7-8, lines 138-149) and Discussion (pp. 12, lines 232-234). The simulated logFC is included in Table S5.

We did not explicitly compute power for the tweedie GLMM and GEE1 models, because our primary point was to emphasize appropriately accounting for the within-person correlation. However, assuming sufficient sample size (independent experimental units) to justify the use of GEE1, these correlations suggest the tweedie GLMM and GEE1 should perform well.

11. Did you run muscat to see if it could recapitulate the statistical properties of the real data as well as your simulation? The countsimQC Bioconductor package can be used to compare multiple count matrices (i.e. simulated data) against the real data to see whether the simulated data recapitulates properties of the real data.

The primary purpose of our manuscript is to emphasize the importance of accounting for within-person correlation when computing differential expression testing with scRNA-seq data. We strongly believe our simulation demonstrates the main point of emphasis for this manuscript. Thus, we feel that comparing our simulation engine to a different simulation engine built for different reasons would distract from the primary purpose of this paper.

REVIEWERS' COMMENTS

Reviewer #1 (Remarks to the Author):

The authors have addressed my comments.

Reviewer #2 (Remarks to the Author):

Again, I appreciate the authors incorporating suggestions from previous rounds of revision. It is reassuring to see that the additional 10x dataset included in Figure 1 shows increased intra-individual correlation compared to inter-individual correlation, although the correlation is notably lower than for the plate-based data. The additional details included in Methods that describe the simulation makes it clearer to follow, and the ranking of the methods with the true log-fold-changes is an interesting addition and provides additional insight into how the methods perform. I was able to access the github repository with the complete link provided.

Regarding the simulation study: the reason I have previously asked about comparisons to other available simulations such as muscat is because it would be good to see how sensitive the various differential expression methods are to the assumptions underlying the simulation that the authors have developed. The authors' simulation is very specific to plate-based data that is on the transcripts-per-million scale, and may not be appropriate for counts generated from more modern droplet-based data, and the bulk of the paper focuses on results obtained from these simulations. It is highly unusual for droplet-based datasets to have large counts and thus large library sizes, in particular protocols that use UMIs will have much smaller counts with library sizes more likely to be closer to 10,000 rather than 1 million. It is not clear how robust the results of the simulation study are and how they would translate to droplet-based count data.

In addition, the paper appears to be incomplete by only including a subset of the methods for the power and sensitivity studies (Figures S5 – S8). An unbiased evaluation would include all methods from Table 1 in these comparisons.

There are some statements that seem to lack evidence in the paper:

- Pg3 line 68: "Aggregation methods are implemented to control for both zero-inflation and within sample correlation, but are conservative and perform poorly in unbalanced situations, which are common in single-cell data." I definitely agree these methods are conservative, but I would not go so far as to say they perform poorly. They correlate just as highly with the true log-fold-changes as the mixed effects model, and are only slightly underpowered in unbalanced scenarios.
- Pg 13 line 253: "The problem of imbalance is most apparent when summing, where differences are primarily driven by the quantity of cells per individual rather than true effects." I see no evidence for this statement in the paper.

Evaluation of rank order correlation: Should the ranking be compared to the $\text{abs}(\log\text{-fold-change})$ rather than $-\log\text{-fold-change}$ (Table S5)? In Methods it is stated that the fold change ranged from 0 – 4. Fold changes < 1 will have negative log-fold-changes (i.e. down-regulated) and fold changes > 1 will have positive log-fold-changes (i.e. up-regulated). Fold changes of exactly 1 indicate no change between the groups. For Figure S5 is the x-axis indeed fold-change or log-fold-change? Why were the negative log-fold-changes ignored?

Division of Public Health Sciences
Department of Biostatistics and Data Science

December 14th, 2020

Dear Reviewers,

We greatly appreciate the work you have put in to reviewing and working to improve our manuscript. Below we provide a point-by-point response to each of your latest comments. All of the changes we made in the manuscript are highlighted in yellow.

Again, we greatly appreciate your insights, time, and efforts.

Sincerely,

Carl D. Langefeld, Ph.D.
Professor, Department of Biostatistics and Data
Science
Division of Public Health Sciences
Wake Forest School of Medicine
Medical Center Boulevard, WC 2321
Winston-Salem, NC 27157-1063

Kip D. Zimmerman, PhD
Molecular Genetics and Genomics
Wake Forest School of Medicine
Medical Center Boulevard, WC 2321
Winston-Salem, NC 27157-1063

Reviewer #1 (Remarks to the Author):

The authors have addressed my comments.

Once again, we thank the reviewer for their work to improve this manuscript. The reviewer's insights have been very valuable in refining the primary message of our work.

Reviewer #2 (Remarks to the Author):

Again, I appreciate the authors incorporating suggestions from previous rounds of revision. It is reassuring to see that the additional 10x dataset included in Figure 1 shows increased intra-individual correlation compared to inter-individual correlation, although the correlation is notably lower than for the plate-based data. The additional details included in Methods that describe the simulation makes it clearer to follow, and the ranking of the methods with the true log-fold-changes is an interesting addition and provides additional insight into how the methods perform. I was able to access the github repository with the complete link provided.

We thank the reviewer as we have sincerely worked to integrate the reviewer's important points while emphasizing the primary focus of the manuscript (i.e., accounting for the within-subject correlation). We believe these suggestions have greatly improved our manuscript.

Regarding the simulation study: the reason I have previously asked about comparisons to other available simulations such as muscat is because it would be good to see how sensitive the various differential expression methods are to the assumptions underlying the simulation that the authors have developed. The authors' simulation is very specific to plate-based data that is on the transcripts-per-million scale, and may not be appropriate for counts generated from more modern droplet-based data, and the bulk of the paper focuses on results obtained from these simulations. It is highly unusual for droplet-based datasets to have large counts and thus large library sizes, in particular protocols that use UMIs will have much smaller counts with library sizes more likely to be closer to 10,000 rather than 1 million. It is not clear how robust the results of the simulation study are and how they would translate to droplet-based count data.

We agree with the reviewer, that our simulation engine (which was initially developed from plate-based data) may simulate single-cell data that will have much larger library sizes than droplet-based count data. However, we emphasize that we did demonstrate the within-sample correlation in droplet-based count data and that is what is of interest in these analyses. The recommendation of applying random effects to account for the within-sample correlation will hold true, as long as the hierarchical correlation structure exists in these data. As we noted in the manuscript, a random effect model can be applied to a range of distributions (e.g., tweedie, negative binomial, zero-inflated negative binomial).

In addition, the paper appears to be incomplete by only including a subset of the methods for the power and sensitivity studies (Figures S5 – S8). An unbiased evaluation would include all methods from Table 1 in these comparisons.

We have updated Supplementary Figure 5 to include evaluations of the sensitivity for all ten methods. However, the power analyses here (Supplementary Figures 6-8) were specifically computed to demonstrate that random effects models are more powerful than “pseudo-bulk” approaches when there is imbalance. We believe adding additional methods that do not account for the within-subject correlation to these plots will distract from the main purpose of these plots. In addition, it is important to recognize that

it is not appropriate to compute power estimates when the type 1 error rate is so grossly inflated. Such estimates of power without complex adjustments to rescale the test statistic to have approximately appropriate type 1 error rate will grossly overestimate power and will mislead readers.

There are some statements that seem to lack evidence in the paper:

• **Pg3 line 68: “Aggregation methods are implemented to control for both zero-inflation and within sample correlation, but are conservative and perform poorly in unbalanced situations, which are common in single-cell data.”** I definitely agree these methods are conservative, but I would not go so far as to say they perform poorly. They correlate just as highly with the true log-fold-changes as the mixed effects model, and are only slightly underpowered in unbalanced scenarios.

We agree and we have removed the phrase “perform poorly”.

• **Pg 13 line 253: “The problem of imbalance is most apparent when summing, where differences are primarily driven by the quantity of cells per individual rather than true effects.”** I see no evidence for this statement in the paper.

This statement was highlighting that “the problem of imbalance is most apparent when summing” because, for example, if one individual had 50 cells and another individual had 5000 cells it is easy to see why summing the expression values (even if every cell had an equal expression value) will lead to differences. However, we see how this statement is misleading and confusing, so we have removed it, because it did not provide additional value to the main premise of this manuscript.

Evaluation of rank order correlation: Should the ranking be compared to the abs(log-fold-change) rather than -log-fold-change (Table S5)?

This is an excellent catch by the reviewer. Thank you. The rankings should absolutely be compared to the absolute value of the $-\log(\text{fold-change})$. We have adjusted this in our text and now report the updated rank-order correlations in Supplementary Table 5.

In Methods it is stated that the fold change ranged from 0 – 4. Fold changes < 1 will have negative log-fold-changes (i.e. down-regulated) and fold changes > 1 will have positive log-fold-changes (i.e. up-regulated). Fold changes of exactly 1 indicate no change between the groups. For Figure S5 is the x-axis indeed fold-change or log-fold-change? Why were the negative log-fold-changes ignored?

The x-axis of Supplementary Figure 5 is indeed fold-change and not the $\log(\text{fold-change})$. The negative $\log(\text{fold-change})$ values were not ignored, they just were not plotted because of redundant information and the positive $\log(\text{fold-change})$ values were more natural to visualize. Sensitivity will be the same for reciprocal fold-changes, like 0.8 and 1.25.